# Optimal blade pitch control for enhanced vertical-axis wind turbine performance

Sébastien Le Fouest[1] & Karen Mulleners [1] ✉

Vertical-axis wind turbines are great candidates to enable wind power extraction in urban and off-shore applications. Currently, concerns around turbine efficiency and structural integrity limit their industrial deployment. Flow control can mitigate these concerns. Here, we experimentally demonstrate the potential of individual blade pitching as a control strategy and explain the flow physics that yields the performance enhancement. We perform automated experiments using a scaled-down turbine model coupled to a genetic algorithm optimiser to identify optimal pitching kinematics at on- and off-design operating conditions. We obtain two sets of optimal pitch profiles that achieve a three-fold increase in power coefficient at both operating conditions compared to the non-actuated turbine and a 77% reduction in structure-threatening load fluctuations at off-design conditions. Based on flow field measurements, we uncover how blade pitching manipulates the flow structures to enhance performance. Our results can aid vertical-axis wind turbines increase their much-needed contribution to our energy needs.

According to the International Energy Agency, the installed wind power capacity should increase 11 times between 2020 and 2050 to meet the global *net-zero emissions by 2050* objective[1]. Wind power is expected to cover up to 31% of the electricity supply by 2050, which is logistically challenging as the overall capacity is limited by the availability of exploitable land[2,3]. The installation of new wind farms alters wind conditions and decreases the performance of existing downwind farms[4,5]. An increase in diversity of wind turbine technology can help mitigate concerns around land use and wake interference[6,7].

Vertical-axis wind turbines provide an attractive design that complements their more ubiquitous horizontal-axis counterparts. By adding vertical-axis turbines to densify existing hortizontal-axis wind turbine farms, the farm's power output is increased by up to an order of magnitude[8–10]. Vertical-axis, or cross-flow, turbines rotate about an axis orthogonal to the incoming flow, which makes them insensitive to wind direction and allows them to prosper in vortex-dominated urban flows[9,10]. They typically operate at lower rotational frequencies, which significantly reduces noise and the risk of collision with avian species[11,12]. Crucial mechanical parts in the drive train can be placed close to the ground. This greatly facilitates maintenance, reduces structural loads, and lowers the centre of mass, which benefits floating off-shore applications[13,14].

The aerodynamic complexity of vertical-axis wind turbines has hampered their industrial development and deployment. The turbine blades encounter varying flow conditions throughout a single turbine rotation, even in a steady wind. When the turbine operates at a low tip-speed ratio $\lambda$, which is the ratio between the blade velocity $\Omega R$, and the wind velocity $U_\infty$, the blades perceive significant amplitude changes in the angle of attack and relative wind velocity (Fig. 1). These varying flow conditions can give rise to unsteady flow separation or dynamic stall[15–18].

In general, dynamic stall refers to the succession of aerodynamic events that occur when an airfoil's angle of attack exceeds its critical static stall angle following a dynamic motion[19]. Early observations of dynamic stall were made by Kramer[20]. Later, it became a topic of interest in helicopter rotor aerodynamics, which motivated a series of investigations using pitching and surging airfoils that revealed the sequence of events that lead to full stall on unsteady airfoils. This sequence consists of the spread of flow reversal over the chord, the formation of a large-scale leading edge stall vortex and associated lift overshoot, followed by the increase of the nose-down pitching moment and vortex shedding[21–24].

More recently, dynamic stall is studied in the context of vertical-axis and cross-flow turbines using computational modelling[25–27], and

[1]Institute of Mechanical Engineering, École Polytechnique Fédérale de Lausanne (EPFL), CH-1015 Lausanne, Switzerland. ✉e-mail: karen.mulleners@epfl.ch

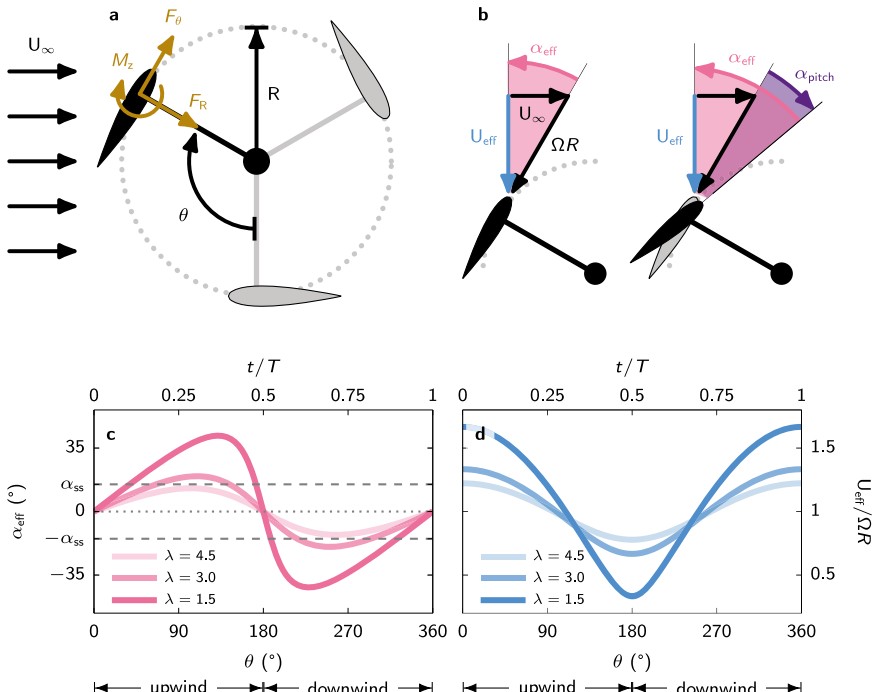

**Fig. 1 | Aerodynamics of a vertical-axis wind turbine blade. a** Schematic representation of an H-type vertical axis turbine with radius $R$ viewed from above. The force direction convention used in this study is represented by the positive direction of the radial force $F_R$, azimuthal force $F_\theta$, and pitching moment around the quarter-chord $M_z$. **b** Zoomed-in view of the velocity triangle at the blade level showing the blade velocity $\Omega R$, the wind velocity $U_\infty$, and the resulting effective velocity $U_{eff}$ as well as the effective angle of attack $\alpha_{eff}$ and pitch angle $\alpha_{pitch}$. **c** Variation of the effective angle of attack $\alpha_{eff}$ as a function of the blade's azimuthal position $\theta$ for different tip-speed ratios. **d** Variation of the effective flow velocity $U_{eff}$ seen by the turbine blade as a function of the blade's azimuthal position $\theta$ for different tip-speed ratios. The amplitude and asymmetry of the variations in effective angle of attack and effective velocity increase with decreasing tip-speed ratio $\lambda$. For tip-speed ratios below $\lambda \approx 3$, the magnitude of the effective angle of attack exceeds the static stall angle of attack $\alpha_{ss}$, indicated with a dashed line in (c).

experiments using particle image velocimetry (PIV)[17,18,28,29]. These investigations show that the power coefficient of vertical axis turbines drops significantly at low tip-speed ratios ($\lambda < 2.5$), where the effective angle of attack increases well above the blade's critical stall angle for extended periods of time (Fig. 1c), leading to the occurrence of deep dynamic stall. High power coefficient are achieved at intermediate tip-speed ratios, around $\lambda \approx 3$[11,27,30]. For high tip-speed ratios ($\lambda > 4$), the effective angle of attack experienced by the turbine blade remains low throughout the turbine's rotation, resulting in low aerodynamic forces and power coefficients. For these reasons, vertical-axis wind turbines typically operate at intermediate tip-speed ratios.

For wind turbine applications, the large-scale vortex shedding and load fluctuations associated with dynamic stall are considered undesirable because they lead to a significant loss in efficiency and load transients that jeopardise the turbine's structural integrity[31–33]. Control strategies at the blade scale include surface actuators, such as plasma actuators[34,35] or blowing and suction slots[36,37]. The goal of these strategies is to locally energise the boundary layer near the surface and delay or prevent separation. High installation and maintenance cost have hampered the commercial deployment of such blade surface flow actuators.

Alternative strategies at the turbine level to control the performance of vertical-axis turbines are intracycle control of the turbine's rotational velocity[12,38] or blade pitching[11,25,39]. These two strategies modify the unsteady blade kinematics within one turbine rotation with the goal to control the overall turbine power. Both methods modify the blade's effective angle of attack to manipulate the severity of the flow separation and the timing of the stall onset and vortex shedding. The stall delay and vortex shedding timescales are largely independent of the rotor frequency and on the order of a few

convective times[40,41]. The duration of the turbine rotation in convective times based on the blade velocity equals $\pi(c/D)^{-1}$ with $c/D$ the chord-to-diameter ratio[42]. Most vertical-axis turbines have a chord-to-diameter ratio below 0.2 and a turbine rotation duration above 15 convective times. The actuation frequency required to control the blade kinematics should be of the same order of magnitude as the turbine's rotational frequency. Surface actuators target the fast evolution of the blade's boundary layer when stall onset is reached and need to operate at frequencies that are orders of magnitude larger than the turbine rotational frequency and the large-scale vortex shedding frequency.

Blade pitching provides the most direct way to modify blade kinematics, but requires more mechanical components than intracycle rotational velocity control. Static and synchronous blade pitching are mechanically simpler than individual blade pitching, but the latter is desirable for its versatility. Here, we demonstrate the potential of individual dynamic blade pitching to enhance the efficiency and maintain the structural integrity of vertical-axis wind turbines across tip-speed ratios using our unique set-up that consist of a scaled-down one bladed instrumented turbine model with dynamic blade pitching capabilities[28]. The turbine's efficiency is conventionally expressed by its power coefficient, which is defined as the ratio of the net mean power generated by the turbine $P_{net}$ and the power carried by the flow crossing the blade's swept area $A_{swept}$:

$$C_P = \frac{P_{net}}{\frac{1}{2}\rho U_\infty^3 A_{swept}} . \qquad (1)$$

The net power $P_{net}$ accounts for the power cost of actuating the turbine blade. More details on the calculation of the power coefficient can be found in the Methods.

## Results

### Optimisation objectives and wind scenarios

Our two optimisation objectives are maximising the power coefficient while minimising the aerodynamic load fluctuations. These objectives are expected to compete in the search for optimal blade kinematics. Optimising an unsteady aerodynamic system in a vast and non-linear design space with competing objectives is challenging for conventional gradient-based methods[43–45]. This challenge motivated the use of alternative optimisation techniques such as evolutionary or genetic algorithms. Genetic algorithms mimic natural evolution by iteratively generating and testing populations, promoting the reproduction of the fittest members through combinations and mutations based on user-selected objectives. Selecting multiple competing objectives will typically give rise to a set of optimal trade-off or Pareto-optimal solutions[46]. Genetic algorithms have repeatedly proved a strong aptitude to identify Pareto-optimal sets in complex multi-objective optimisations of unsteady aerodynamic systems over the past three decades[43,47–50]. This aptitude stems from their inherent resilience to getting stuck in local minima, their ability to identify multiple optimal solution per generation, and their relative simplicity and interpretability[46].

We couple the turbine model to a genetic algorithm-based optimiser and perform series of automated experiments to determine optimal pitching kinematics for two wind scenarios that are typically encountered by industrial wind turbines. For a given wind turbine geometry, there is an optimal tip-speed ratio at which the turbine reaches its maximum power coefficient[51]. An industrial wind turbine will tune its rotational frequency to operate at the optimal tip-speed ratio for a given wind speed. Structural constraints limit the maximum rotational frequency. Once the turbine reaches its maximum rotational frequency, a further increase in wind speed will decrease the tip-speed ratio.

The first scenario we consider relates to this off-design condition where excessively high wind speeds caused the turbine to operate at a tip-speed ratio below its optimal value. Low tip-speed ratios lead to prohibitively high and unsteady loads acting on the turbine blades[52]. This high wind scenario threatens the turbine's structural integrity and is associated with a loss of efficiency. We performed a tip-speed ratio sweep and determined that $\lambda = 1.5$ is representative of a low tip-speed ratio for our turbine geometry (Supplementary Fig. 1).

The second scenario deals with on-design conditions where the non-actuated wind turbine is operating at its optimal tip-speed ratio, which occurs at $\lambda = 3.2$ for our turbine geometry (Supplementary Fig. 1).

We assess the potential of dynamic blade pitching to ensure safe and efficient turbine operation for both scenarios. The optimisation objectives are increasing the turbine's mean power coefficient and reducing the blade pitching moment's standard deviation. The latter objective is used to quantify the intensity of undesirable load fluctuations related to flow separation from wind turbine blades[28].

The proposed pitching kinematics are a sum of sine waves with three harmonics of the turbine rotational frequency $\Omega$:

$$\alpha_{\text{pitch}}(t) = A_0 + \sum_{n=1}^{3} A_n \sin(n\Omega t + \theta_n) \quad , \tag{2}$$

where $A_0$ is a fixed angle offset, $A_n$ is the amplitude and $\theta_n$ the phase shift of the $n^{th}$ harmonic. This leads to a total of seven optimisation parameters. We use harmonics of the turbine frequency to enforce periodicity and use only the first three harmonics, similar to what is done in[12]. The first three harmonics allow for sufficiently large local pitching gradients and phase shifts to change the extreme values of the effective angle of attack and when it exceeds its critical limits without introducing higher frequency vibrations.

The experimental set-up and the optimisation routine are summarised in Fig. 2 and discussed in detail in the Methods. A comprehensive description of the selected parameters, constraints, and settings of the genetic algorithm is also given in the Methods.

### Optimisation of pitching kinematics for off-design operation

The results of the bi-objective blade pitching optimisation for the turbine operating at an off-design tip-speed ratio of $\lambda = 1.5$ are summarised in Fig. 3. The performance of all 1800 tested individuals during the optimisation are presented in Fig. 3a in terms of their mean power coefficient, normalised by the mean power coefficient of the non-actuated turbine, and the standard deviation of the pitching moment, also normalised by the standard deviation of the non-actuated turbine. The best-fit individuals that form a Pareto front are coloured corresponding to their relative mean power coefficient. An individual is considered part of the Pareto front or Pareto-optimal when no other individual scored better for both objectives. We further analyse the Pareto-optimal kinematics to highlight their common features and traits.

The optimal individuals improve their power coefficient by a factor 2.5 to 3.2 and reduce the pitching moment standard deviation by 60% to 77% (Fig. 3a). Overall, both objectives can be achieved hand in hand and the remaining trade-off between increasing performance and reducing fluctuations is tied to subtle changes in the pitching kinematics.

The Pareto-optimal pitching kinematics all execute an outward pitching manoeuvre during the upwind phase, followed by inward pitching manoeuvre during the downwind phase (Fig. 3b). The initial outward pitching manoeuvre serves to reduce the blade's effective angle of attack, delay the moment the critical static stall angle is exceeded, and reduce the maximum effective angle of attack (Fig. 3c, d). The blade pitching kinematics that yield the highest power coefficients have a higher amplitude of the mean offset angle (Fig. 3e) and have a higher contribution of the third order terms from the sine-series described by equation (2) (Fig. 3f). The highest power coefficient kinematics with a large amplitude of $A_3$ also yield the lowest reduction in the pitching moment fluctuations. The variations for the amplitudes of all sine terms are presented in Supplementary Fig. 2. These more complicated kinematics perform well in our controlled laboratory environment, but for general applications, we would recommend using the simpler kinematics that are likely to be more robust and further reduce the pitching moment standard deviation.

The inward pitching manoeuvre during the downwind phase serves to decrease the magnitude of the blade's effective angle of attack below the critical stall angle between $t/T = 0.70$ and $t/T = 0.78$ for all Pareto-optimal solutions (Fig. 3c). The magnitude of the effective angle of attack of the non-actuated turbine blade remains above the critical limit during most of the downwind ($|\alpha_{\text{eff}}| > \alpha_{\text{ss}}$ from $t/T = 0.52$ to $t/T = 0.90$) which prevents the turbine from creating significant power during the downwind phase (Fig. 4). The earlier recovery in the Pareto-optimal blade pitching cases allows for a second region of substantial power production in the second half of the downwind phase (Fig. 4).

To uncover the impact of the blade kinematics on the power coefficient, we analyse the combined results of the time-resolved power coefficient and flow field measurements in Fig. 4a, b for the non-actuated and an exemplary Pareto-optimal pitching blade (indicated by the coloured pentagon symbol in Fig. 3a). The non-actuated case is characterised by the occurrence of deep dynamic stall (Fig. 4a). As the blade climbs upwind, a leading-edge vortex forms, increases the leading edge suction, and allows for a peak in power production around $\theta = 80°$. When the vortex separates from the blade before the end of the upwind phase, the blade forces and the power coefficient collapse. In the downwind half of the cycle, the outward side of the airfoil becomes the suction side, and the flow remains mostly

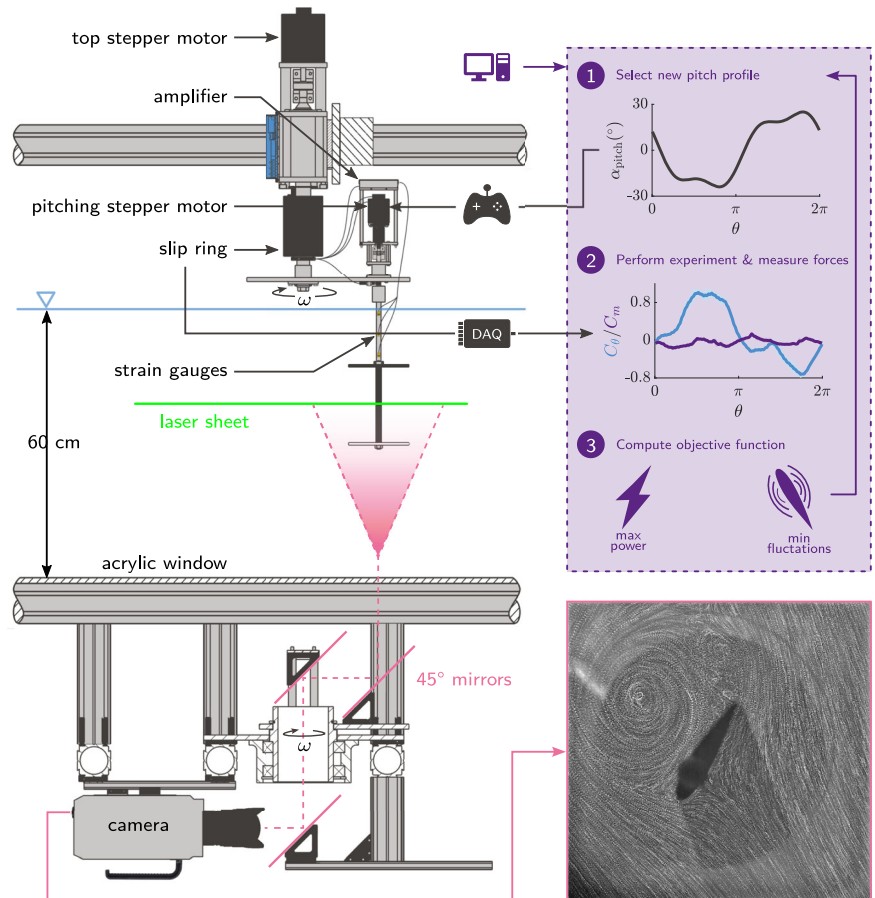

**Fig. 2 | Experimental apparatus and methods.** Cross-section view of the experimental set-up including the wind turbine model, the light sheet, the rotating mirror system, and high-speed camera for particle image velocimetry (PIV). The optimisation of the blade kinematics fully relies on the blade force measurements. The routine followed by the genetic algorithm for each individual is outlined in the top right. An idea of the camera's field of view is shown in the bottom right by the maximum intensity image over a time series of 50 images taken during the formation of a large dynamic stall vortex.

separated due to the high magnitude of the effective angle of attack. This leads to limited power extraction during the downwind phase.

The Pareto-optimal pitching kinematics begin the upwind phase by pitching the blade outwards, reducing the blade's effective angle of attack. This manoeuvre delays flow separation and redirects the aerodynamic force towards the direction tangential to the blade's path (Fig. 4b). A coherent stall vortex is not observed until the end of the upwind phase, and the vortex is significantly smaller than in the non-actuated case. The effective delay of the upwind dynamic stall vortex formation allows the blade to obtain a positive power coefficient during the entire upwind phase. When the blade enters the downwind phase, the blade executes a rapid inward pitch manoeuvre, forcing the stall vortex to shed (Fig. 4b, $180° < \theta < 225°$). By controlling and delaying the timing of the vortex shedding, the stall vortex is now shed towards the side of the rotor to avoid blade-vortex interactions and associated load fluctuations. The absence of blade-vortex interactions combined with the early drop in the magnitude of the effective angle of attack promotes flow reattachment and the occurrence of a prominent second region of power extraction during the downwind phase. The potential to promote flow reattachment by a fast pitching manoeuvre has been demonstrated previously by Prangemeier[53] for plunging airfoils.

All Pareto-optimal kinematics show the same key features that explain their improved performance at off-design operation. An initial outward pitching manoeuvre delays the onset of the upwind dynamic stall and redirects the aerodynamic force forwards during the upwind phase. The timing of stall onset $\theta^*$, identified as the moment when the power coefficient drops below zero after the upwind power generation phase, is delayed from $\theta^* = 125°$ ($t/T = 0.35$) in the non-actuated case to $180° < \theta^* < 190°$ ($0.50 < t/T < 0.53$) for the Pareto-optimal solutions (Fig. 4c). The subsequent inward pitching motion controls the timing and the direction of the shedding of the stall vortex to avoid blade-vortex interactions and allows for early flow reattachment. These combined effects substantially increase the power generated during the downwind phase (Fig. 4d). All Pareto-optimal solutions reach approximately the same average power coefficient of 0.37 over the upwind phase, which is an improvement by a factor of 2.8 with respect to the non-actuated case. The differences between the Pareto-optimal kinematics mainly affect the downwind phase. Overall, blade pitching is a highly effective solution to keep wind turbines safe while boosting their performance during off-design operation.

**Optimisation of pitching kinematics for on-design operation**
Ideally, vertical-axis wind turbines operate most of the time at on-design tip-speed ratios. To confirm that individual blade pitching is still worth the investment under ideal operating condition, we also conducted the optimisation experiments at the tip-speed ratio of $\lambda = 3.2$, where our non-actuated experimental turbine model reaches its highest power coefficient. The results are summarised in Fig. 5.

The Pareto-front has two branches, a lower branch where $\sigma(M_z) \leq \sigma(M_{z,na})$, and a higher branch where $\sigma(M_z) > \sigma(M_{z,na})$ (Fig. 5a). The Pareto-optimal individuals in the higher branch improve their power coefficient by a factor of 2.4 to 2.9 at the expense of increasing their load fluctuations up to 186%. The level of load fluctuations for the

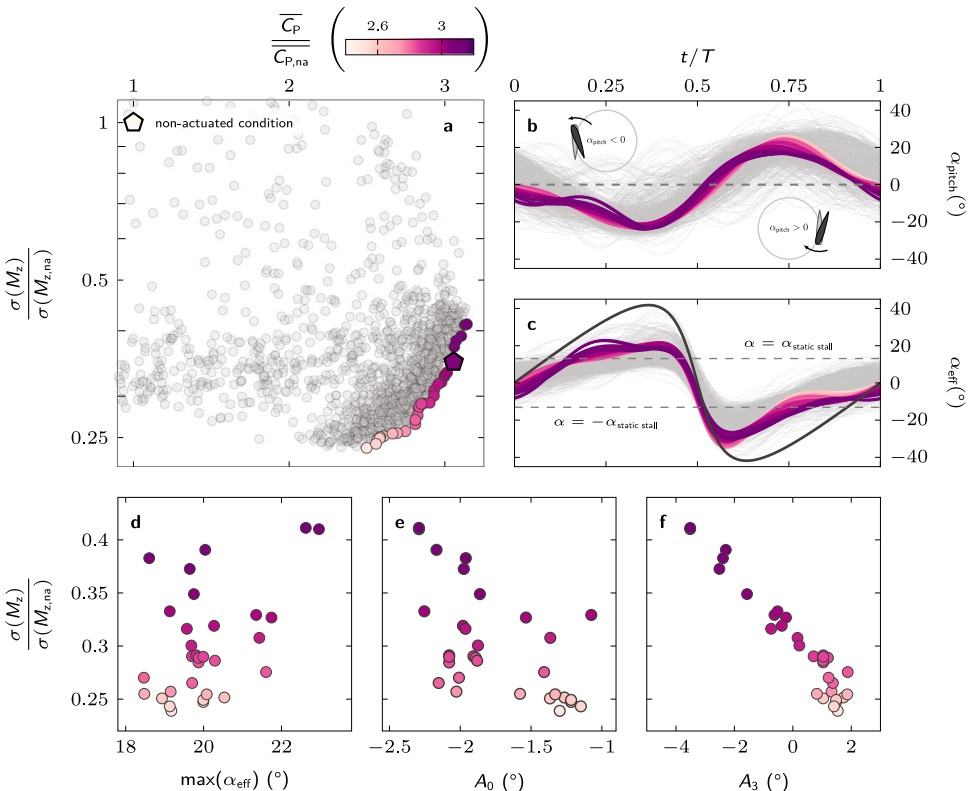

**Fig. 3 | Optimal pitching kinematics for off-design operation. a** Normalised mean standard deviation of the pitching moment versus normalised mean power coefficient for all tested individuals. The individuals that form a Pareto front are coloured corresponding to their normalised mean power coefficient. **b** Temporal evolution of the blade pitch angle for all individuals (light grey) and Pareto-optimal individuals (coloured). A negative pitch angle corresponds to an outward rotation of the blade's leading edge and vice versa. The line colours refer to the normalised mean power coefficient. **c** Temporal evolution of the effective angle of attack for the non-actuated turbine at this optimal tip-speed ratio is low and is not a concern for the safe operation of the turbine. If we do not want to risk increasing the level of load fluctuations, we can opt for solutions on the lower branch of the Pareto front, which can still improve their power coefficient up to a factor of 2.3 while maintaining the same load fluctuations as the non-actuated case.

the non-actuated individual (black), all tested individuals (light grey) and Pareto-optimal individuals (coloured). **d** Maximum value of the effective angle of attack reached during the upwind phase, for the different Pareto-optimal solutions sorted according to their relative fluctuation level. **e** Mean offset angle ($A_0$) and (**f**) amplitude of the third harmonic ($A_3$) of the Pareto-optimal pitching kinematics. (Source data are publicly available on Zenodo https://doi.org/10.5281/zenodo.10776724).

The pitching kinematics of the individuals on the lower branch of the Pareto front resemble simple low amplitude sinusoidal motions at the rotor frequency (Fig. 5b). This leads to a reduction in the gradient of the effective angle of attack during extended parts of the cycle (Fig. 5c) and a maximum effective angle of attack around 5°, which is well below the critical static stall limit (Fig. 5d). All Pareto-optimal kinematics have higher offset angle magnitudes |$A_0$| compared to the optimal pitching solutions at off-design operation (Fig. 5e). The highest power-extracting pitching kinematics have a higher contribution of the higher harmonics (Fig. 5f).

The development of the power coefficient and the vorticity fields for the non-actuated blade and an exemplary Pareto-optimal pitching blade at $\lambda = 3.2$ (indicated by the coloured pentagon symbol in Fig. 5a) are presented in Fig. 6. The non-actuated blade is characterised by the occurrence of light dynamic stall (Fig. 6a). Compared to the situation at lower tip-speed ratio, the upwind separation region does not evolve into a coherent vortex. The moderate effective angle of attack and absence of deep dynamic stall allow the turbine at $\lambda = 3.2$ to generate a positive power coefficient during most of the upwind phase. During the downwind phase, the flow reattaches. The fluid dynamic force acting on the non-actuated blade is significantly lower during the downwind than during the upwind phase.

The main benefit of blade pitching during off-design operation is the efficient delay and mitigation of deep stall and a substantial increase in the power production during the upwind phase. For on-design operation, deep stall is absent and the effect of blade pitching is most prominent during the downwind phase (Fig. 6b, d). The optimal pitching kinematics on the lower branch of the Pareto front achieve a reduction in the load fluctuations during the downwind phase by reducing the effective angle of attack magnitude. The magnitude of the effective angle of attack minimum for these individuals barely exceeds the critical static stall angle (indicated by the colour of the triangles in Fig. 65c). The minimum effective angle of attack is reached within the first third of the downwind phase (triangles in Fig.6 5c) and the magnitude of the effective angle of attack has dropped below the static stall limit in the first half of the downwind phase (circles in Fig. 65c). This strategy reduces the load fluctuations, but it limits the improvement in power production during the downwind phase. The kinematics that yield the strongest reduction in the load fluctuations even have negative power production during the downwind phase (Fig. 6d).

The optimal pitching kinematics on the upper branch of the Pareto front reach a magnitude of the minimum effective angle of attack up to 9° above the static stall value (indicated by the colour of the triangles in Fig.6 5c). The time at which they reach the minimum effective angle converges to a constant value around $\theta = 240°$, and the time at which their angle of attack returns below the static stall angle

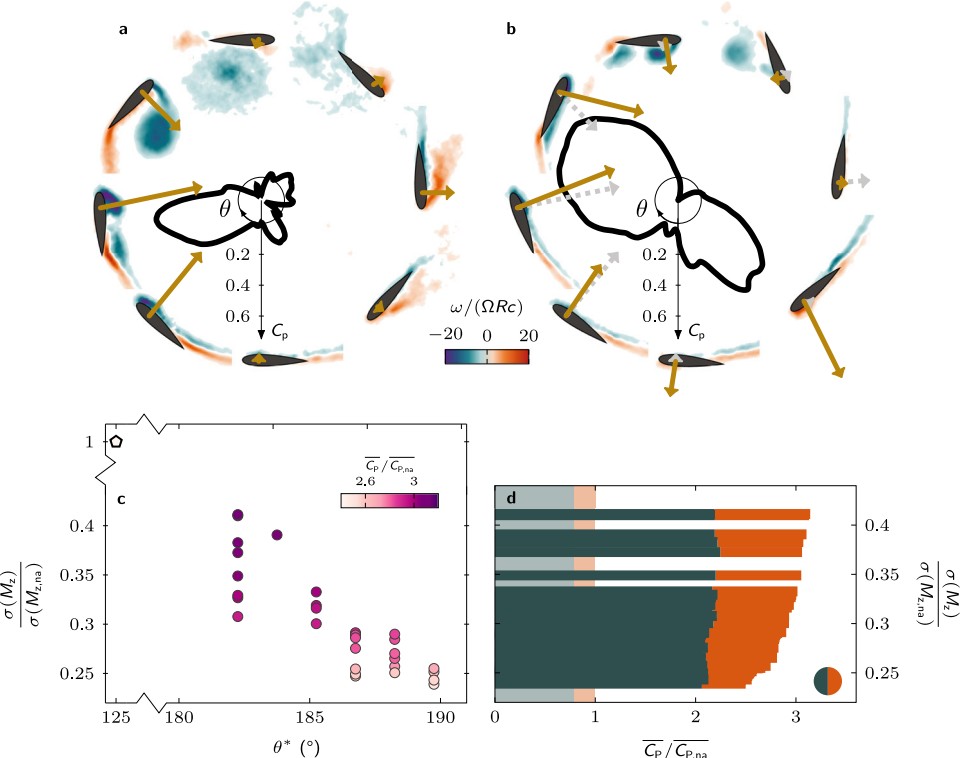

**Fig. 4 | Physical insights into performance enhancement from optimal pitching kinematics at off-design operation.** Polar plot comparison of the phase-averaged power coefficient at tip-speed ratio $\lambda = 1.5$ for both **a** the non-actuated case and **b** exemplary optimal kinematics indicated by the coloured pentagon marker in Fig. 3a. Phase-averaged normalised vorticity ($\vec{\omega} = \nabla \times \vec{u}$) fields are shown at eight equally spaced azimuthal positions ($\theta = [0°: 45°: 360°]$) to illustrate the development of flow structures for both cases. The arrows indicate the size and direction of the force acting on the blade. The dashed arrows in **b** indicate the size and direction of the blade force in the non-actuated case for comparison. **c** Timing of the dynamic stall onset $\theta^*$ for the different Pareto-optimal solutions sorted according to their relative fluctuation level. **d** Comparison of the power extracted during the upwind and downwind parts of the cycle for the Pareto-optimal solutions sorted according to their relative fluctuation level. The shading in the background indicates the values for the non-actuated case. (Source data are publicly available on Zenodo https://doi.org/10.5281/zenodo.10776724).

converges to $\theta = 270°$ (circles in Fig.6 5c). The non-actuated blades operate at effective angles of attack beyond the static stall angle for a longer duration. The magnitude of the effective angle of attack of the non-actuated blades only reaches safe values at $\theta = 304°$, which is after two thirds of the downwind phase. Optimal pitching kinematics promote early flow reattachment by accelerating when their effective angle of attack crosses the static stall angle limit. The reattached flow allows for the formation of a downwind leading edge vortex, yielding an additional power production region in the second part of the downwind phase for the kinematics on the upper branch of the Pareto front.

The positive effect of flow reattachment early in the downwind phase is clearly visible when comparing the downwind snapshots for the non-actuated and actuated case in Fig. 6a, b. The boundary layer in the non-actuated case is not yet fully reattached at $\theta = 270°$ which is evidenced by the positive vorticity in the wake and little to no force is generated under these circumstances. In the actuated case, shown in Fig. 6b, no signs of flow separation on the outer side are visible and little to no positive vorticity is present in the wake for $180° < \theta < 270°$. The absence of flow separation combined with high magnitude effective angles of attack close to the stall limit lead to the accumulation of positive vorticity near the leading edge of the actuated airfoil at $\theta = 315°$ and a substantially larger force vector than in the non-actuated case even though the instantaneous geometric conditions are similar at $\theta = 315°$. The unsteady force response is strongly affected by the past evolution of the flow and not solely governed by the instantaneous angles of attack and effective flow velocities. The accurate prediction of these unsteady history effects is still a major challenge for

theoretical modelling but our data-driven approach is able to find dynamic pitch angle solutions that exploit them.

All Pareto-optimal kinematics yield a similar improvement of the average power coefficient during the upwind phase and most of the variations are observed during the downwind phase (Fig. 6d). The increase in the downwind power coefficient goes together with an increase in the fluctuation levels. The kinematics that lead to the lowest fluctuations do not extract power during the downwind phase. The higher power extracting kinematics achieve an even higher power coefficient during the downwind than during the upwind phase. These results demonstrate that even at on-design tip-speed ratios, blade pitching offers the opportunity to further improve the power extraction without compromising the structural resilience of the wind turbines.

## Discussion

In this study, we demonstrate that individual blade pitching is an effective control strategy to improve the performance of vertical-axis wind turbines across tip-speed ratios. A family of optimal blade pitching kinematics are derived with an in-situ experimental optimisation using a reduced-scale turbine model coupled to a genetic algorithm. In the controlled laboratory environment, optimal dynamic blade pitching can achieve a threefold increase in turbine power coefficient at both on- and off-design tip-speed ratios.

The two blade pitching manoeuvres that are key to success are an outward pitch manoeuvre during the upwind phase and an inward pitch manoeuvre during the downwind phase. The outward pitch manoeuvre limits the overshoot of the effective angle of attack past

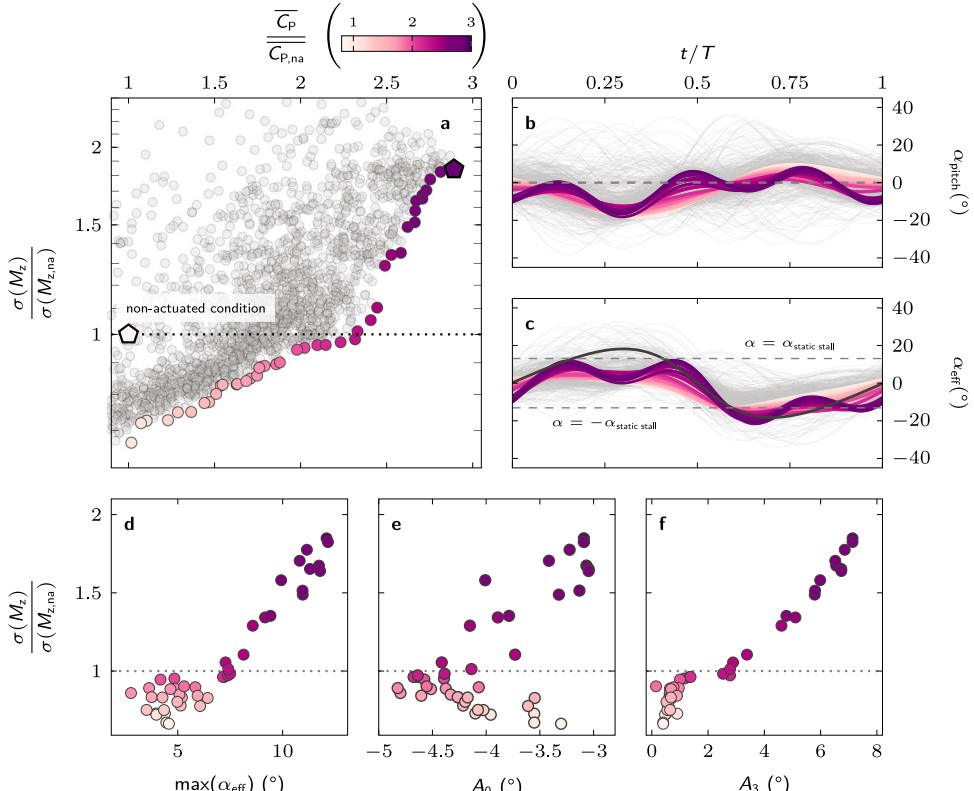

**Fig. 5 | Optimal pitching kinematics for on-design operation. a** Normalised mean standard deviation of the pitching moment versus normalised mean power coefficient for all tested individuals. The individuals that form a Pareto front are coloured corresponding to their normalised mean power coefficient. **b** Temporal evolution of the blade pitch angle for all individuals (light grey) and Pareto-optimal individuals (coloured). A negative pitch angle corresponds to an outward rotation of the blade's leading edge and vice versa. The line colours refer to the normalised mean power coefficient. **c** Temporal evolution of the effective angle of attack for the non-actuated individual (black), all tested individuals (light grey) and Pareto-optimal individuals (coloured). **d** Maximum value of the effective angle of attack reached during the upwind phase, for the different Pareto-optimal solutions sorted according to their relative fluctuation level. **e** Mean offset angle ($A_0$) and **f** amplitude of the third harmonic ($A_3$) of the Pareto-optimal pitching kinematics. (Source data are publicly available on Zenodo https://doi.org/10.5281/zenodo.10776724).

the critical stall angle and reorients the blade force towards the tangential direction to increase power extraction during the upwind phase. At off-design tip-speed ratios, this manoeuvre also delays massive flow separation and perfectly times the dynamic stall vortex shedding. By shedding the vortex toward the side at the beginning of the downwind phase, we avoid undesirable vortex-blade interactions during the rest of the downwind phase, which further aids to increase the downwind efficiency and reduce load fluctuations. The downwind inward pitch manoeuvre causes the magnitude of the effective angle of attack to quickly drop and remain near the value of the critical stall angle to optimise the power extraction during the downwind phase. This manoeuvre promotes flow reattachment and enables the formation of a second leading edge vortex, yielding a second power production region in the downwind phase. Vertical-axis wind turbines generally extract little to no power during the downwind phase and the improvements achieved here are transformative gains in efficiency. Deeper analysis on the influence of optimal individual blade pitching on the aerodynamic properties of flow structures forming on the turbine blade, and a comparison of these properties with sub-optimal blade pitching kinematics, are further work.

A necessary and desirable next step is to test this control mechanism at larger scale. Our experiments are conducted with a motor-controlled, single-bladed reduced-scale turbine model, operated at a blade Reynolds number of $Re_c = (\rho\Omega Rc)/\mu = 50000$, where $\rho$ is the density and $\mu$ the dynamic viscosity of water. The turbine blade is lightweight and has a small aspect ratio such that the required moment to pitch the blade was two orders of magnitude smaller than

the torque experienced by the turbine's central shaft for Pareto-optimal individuals. An industrial wind turbine blade would have greater actuation costs, potentially giving an edge to low amplitude pitching kinematics. The motor-controlled turbine is deemed suitable to demonstrate the working principle of dynamic blade pitching and estimate its potential[54]. The primary mechanisms responsible for the success of dynamic blade pitching demonstrated here are linked to the blade-level physics and are likely to be robust to the presence of a generator and the extension to a multi-blade turbine. Generally, the power curves of single-blade and multiple-blade units of conventional cross-flow turbines show reasonable overlap, as the majority of the power is extracted during the upwind phase, where the effect of additional blades is minor[12]. Our optimal pitching kinematics substantially increase the power contribution during the downwind phase and future validation is necessary to quantify the improvement for multi-blade turbines operating at full-scale Reynolds numbers and with increased levels of turbulence intensity. The dynamic stall delay for pitching airfoils in clean inflow conditions is largely independent of the Reynolds number[55], and appears to be more strongly affected by the free-stream turbulence[56]. Increased levels of turbulence intensity can improve the performance of small-scale vertical-axis wind turbines that operate at lower Reynolds numbers by up to 20%, but the effect becomes negligible at larger operational Reynolds numbers[57,58]. More detailed studies on the effect of inflow turbulence on the performance of the wind turbines and the robustness of the individual blade pitch control are desirable.

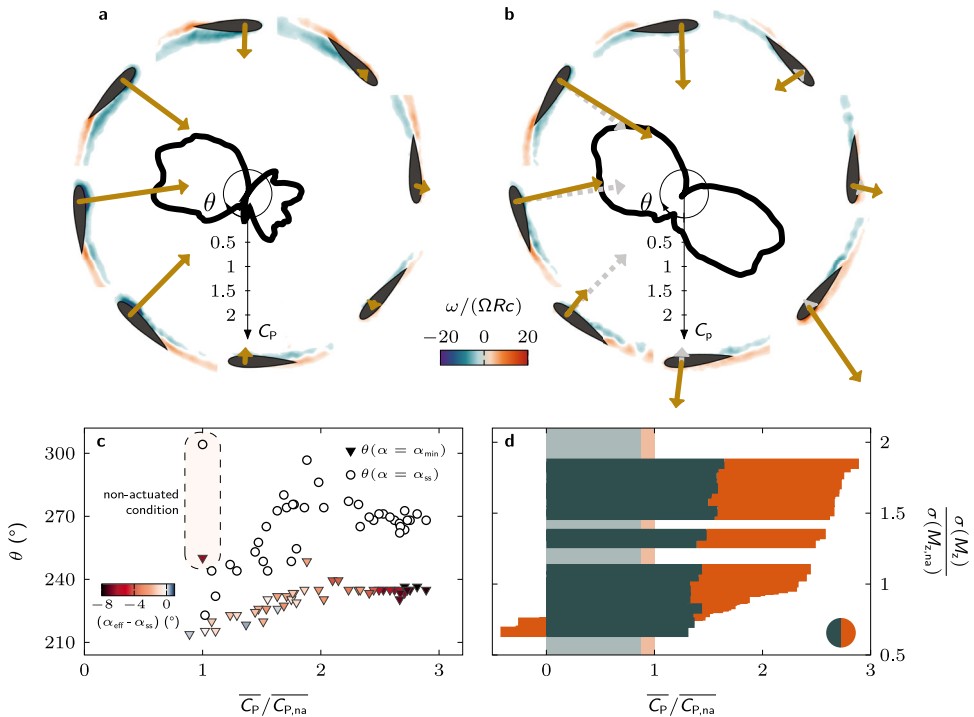

**Fig. 6 | Physical insights into performance enhancement from optimal pitching kinematics at on-design operation.** Polar plot comparison of the phase-averaged power coefficient at tip-speed ratio $\lambda = 3.2$ for both (**a**) the non-actuated case and (**b**) exemplary optimal kinematics indicated by the coloured pentagon marker in Fig. 5a. Phase-averaged normalised vorticity fields are shown at eight equally spaced azimuthal positions ($\theta = [0°: 45°: 360°]$) to illustrate the development of flow structures for both individuals. The arrows indicate the size and direction of the force acting on the blade. The dashed arrows in (**b**) indicate the size and direction of the blade force in the non-actuated case for comparison. **c** Phase angle at which the minimum effective angle of attack is reach during the downwind part of the rotation (triangles) and phase angle at which the magnitude of the effective angle of attack drops below the critical static stall angle (circles) for the non-actuated and Pareto-optimal cases as a function of the normalised power coefficient. The marker colours indicate the effective angle of attack values relative to the static stall angle at the respective phases. **d** Comparison of the power extracted during the upwind and downwind parts of the cycle for the Pareto-optimal solutions sorted according to their relative fluctuation level. The shading in the background indicates the values for the non-actuated case. (Source data are publicly available on Zenodo https://doi.org/10.5281/zenodo.10776724).

Future studies could also invest in developing closed-loop control algorithms to increase the flexibility and adaptability of the individual blade pitch control for unsteady and turbulent inflow conditions and for perturbations caused by the wakes of additional blades in a multi-bladed turbine. Closed-loop control attempts could also explore alternative control objectives for more diverse application scenarios. Instead of striving for the maximum power coefficient, an optimisation that minimises the streamwise force experienced by the turbine as a whole could be desirable, as high streamwise forces lead to heavy stress on the turbine tower and are associated with large momentum deficits in the wake. Blade pitching could also aim for optimal wake recovery or wake steering in wind farm configurations. Complementary efforts are desirable to improve our modelling capabilities of the unsteady blade-level loads and the turbine-level performance to better predict the impact of blade pitching and guide future turbine and wind farm designs. Full-scale vertical-axis wind turbines with a 750 kW rated power and blade pitching capabilities are already available on the market. Dynamic blade pitching is a realistic and affordable mechanism to achieve transformative gains in efficiency and robustness of vertical-axis wind turbines.

## Methods
### Flow conditions
Experiments were conducted in a recirculating water channel with a test section of $0.6\,\text{m} \times 0.6\,\text{m} \times 3\,\text{m}$ and a maximum flow velocity of $1\,\text{ms}^{-1}$. The flow velocity, $U_\infty$, was set using a propeller-based anemometer, to $U_\infty = 0.56\,\text{ms}^{-1}$ for the tip-speed ratio $\lambda = 1.5$ case and to $U_\infty = 0.26\,\text{ms}^{-1}$ for $\lambda = 3.2$. The test section had transparent acrylic walls

that were bound by a metallic frame, providing optimal optical access for velocity field measurements.

The variation of the effective angle of attack, defined as the angle between the blade's chord and the effective velocity vector, as a function of the blade's azimuthal position and the tip-speed ratio $\lambda$ is derived from trigonometry:

$$\alpha_{\text{eff}}(\theta) = \tan^{-1}\left(\frac{\sin\theta}{\lambda + \cos\theta}\right) - \alpha_{\text{pitch}}(\theta). \tag{3}$$

By convention, the effective angle of attack is positive when the effective flow velocity hits the blade on the surface facing outward and vice versa. The blade's pitch angle $\alpha_{\text{pitch}}$ is defined as the angle between the blade's chord and the tangent line to the blade's path. By convention, the pitch angle is positive when the leading edge of the blade is rotated inward and vice versa.

The effective flow velocity $U_{\text{eff}}$ seen by the turbine blade can also be expressed relative to the incoming wind speed with respect to the blade's azimuthal position:

$$U_{\text{eff}}(\theta) = U_\infty \sqrt{1 + 2\lambda\cos\theta + \lambda^2}. \tag{4}$$

### Turbine model
We use a scaled-down model of a single-blade H-type Darrieus wind turbine, mounted in the centre of the water channel's test section. The turbine diameter $D$ was at 30 cm. The turbine blade had a NACA0018

profile with a span of $s = 15$ cm and a chord of $c = 6$ cm, yielding a chord-to-diameter ratio of $c/D = 0.2$. The turbine blade was 3D printed using photosensitive polymer resin (Formlabs Form 2 stereolithography), sanded with fine P180 grit paper and covered with black paint.

The turbine's compact geometry allowed for a relatively small blockage ratio of 12.5%, based on the ratio of the blade's frontal swept area and the water channel's cross section. The blade is held by a cantilevered shaft such that there is no central strut interference with the flow. Two circular splitter plates were placed on the sides of the blade to reduce tip effects. At low tip-speed ratio ($\lambda < 2.5$), the effective blockage was closer to 2.5%, which is the blockage ratio calculated based on the ratio of the blade's surface area to the water channel cross-sectional area. A 2.5 chord length distance to the water channel's side walls was respected at all times. Even though a blockage of 12.5% is likely to enhance the power coefficient, the flow physics and timescales remain comparable to that of an unconfined turbine as demonstrated by an investigation on a single-bladed wind turbine and a 32% blockage ratio[59]. We mitigate the power coefficient enhancement by focusing our analysis on the improvement achieved by the actuation relative to the non-actuated turbine.

The turbine model was driven by a NEMA 34 stepper motor with a $0.05°$ resolution for the angular position. The rotational speed was at $\Omega = 5.6$ rad s$^{-1}$, yielding a constant chord-based Reynolds number based on the blade velocity of $Re_c = (\rho \Omega R c)/\mu = 50000$. An alternative definition of the Reynolds number is used in wind-farm level investigations and is obtained from the free-stream velocity and turbine diameter: $Re_D = (\rho U_\infty D)/\mu$. Here, the turbine model is operating between $Re_D = 168000$ for $\lambda = 1.5$ and $Re_D = 78750$ for $\lambda = 3.2$. The blade was connected to a separate direct-drive stepper motor with a 1:12 gear reduction yielding a high torque mechanism with a $0.0015°$ angular resolution. All motors were individually controlled and synchronized using a three-thread motion controller (GALIL-4080).

## Force measurements

We instrumented the blade shaft with twenty strain gauges forming five full Wheatstone bridge channels to record unsteady aerodynamic loads. The design, calibration, and error quantification of this in-house load cell were detailed in[28]. We recorded shear-forces applied at the blade's mid-span in the radial $F_R$ and azimuthal $F_\theta$ direction, and the pitching moment about the blade's quarter-chord $M_Z$ (Fig. 1a). For each experiment, the wind turbine model started at rest with the blade facing the incoming flow.

The turbine blade was accelerated to its prescribed rotational speed. After reaching the target rotational speed, we waited for five full turbine rotations before starting the load recordings. Aerodynamic forces acting on the turbine blade were recorded at 1000 Hz for 25 rotations using a data acquisition unit (National Instruments, NI 9205), then the blade was brought to rest. After 25 rotations, the mean value of our two objective functions have a normalised root-mean squared error below 1% (Supplementary Fig. 3).

The centripetal inertial force resulting from the turbine's rotation was measured by operating the wind turbine in air. The added drag from the two splitter plates was measured for all investigated tip-speed ratios by operating the wind turbine without the blade, where the two splitter plates were held by a small cylinder. The influence of the centripetal inertial force and the splitter plate drag force were subtracted from the raw measurement data and only the remaining aerodynamic forces acting on the turbine blade were presented. A description of the measurement and modelling of the non-aerodynamic forces was given in[28]. The presented force data was filtered using a second-order low-pass filter with the cut-off frequency at 30 Hz.

## Power calculation

The power generated by the turbine blade is proportional to the aerodynamic force component that is tangential to the blade. Here, we wish to compute the instantaneous net power $P_{net}$ generated by the turbine blade by accounting for the power expense related to the motor actuation:

$$P_{net}(\theta) = \underbrace{F_\theta(\theta)R\Omega}_{\substack{\text{power generated} \\ \text{by the blade}}} - \underbrace{M_Z(\theta)\dot{\alpha}(\theta)}_{\substack{\text{power consumed} \\ \text{by the actuation}}} \qquad (5)$$

where $F_\theta$ and $M_Z$ are the tangential force and pitching moment experienced by the turbine blade, and $\Omega$ and $\dot{\alpha}$ are the turbine's rotational frequency and the blade's pitch rate (Fig. 1). The power coefficient is calculated using equation (1) based on the mean net power generated by the turbine blade throughout 25 rotations. This idealised definition of the power does not account for any mechanical or electronic losses that would occur on a real wind turbine between the turbine blade and the generator's output. Given that were are interested in the relative improvement of the power performance of the turbine, we compare idealised power coefficients across experiments, both actuated and non-actuated.

## Genetic algorithm

We implemented an optimisation framework using the multi-objective genetic algorithm optimiser from the MATLAB global optimization toolbox[60] to obtain a set of optimal blade pitching kinematics. The multi-objective optimisation problem tackled here can be defined mathematically as follows:

$$\text{minimise } \vec{f}(\vec{x}) := \left[ C_P(\vec{x}), \sigma(M_Z(\vec{x})) \right] \qquad (6)$$

subject to

$$g_i(\vec{x}) \le 0, i = 1, 2, \ldots, m \qquad (7)$$

where $\vec{x} = [A_0, A_1, A_2, A_3, \theta_1, \theta_2, \theta_3]^T$ is the parameter vector used to parametrise the blade pitching kinematics (equation (2)). The parameter search space is typically bound by constraint functions $g_i(\vec{x})$. We are minimising two objective functions in parallel: the power coefficient $C_P$ (equations (1) and (5)) and the pitching moment standard deviation $\sigma(M_Z)$. The pitching moment standard deviation was shown to be an excellent proxy for the intensity of undesirable aerodynamic load fluctuations related to flow separation in previous work from the authors[28].

The allowed pitching kinematics were described by a sum of sine waves with three harmonics of the turbine rotational frequency and a fixed angle offset (equation (2)), yielding a total of seven optimisation parameters. The seven parameters are bounded by the values listed in Table 1. The bounds were selected to give the algorithm an extensive search domain but excluding known sub-optimal regions that may increase the convergence time. For instance, the $A_1$ term was constrained to negative values to prevent pitch profiles that lead to

**Table 1 | Lower and upper bounds (LB and UB) that constrained the parameter space for the generation of pitching profile using equation (2)**

|  | $A_0$ | $A_1$ | $A_2$ | $A_3$ | $\theta_1$ | $\theta_2$ | $\theta_3$ |
|---|---|---|---|---|---|---|---|
| LB$_{\lambda = 1.5}$ | −10° | −31° | −15° | −8° | −45° | −60° | −60° |
| UB$_{\lambda = 1.5}$ | 10° | 0° | 15° | 8° | 45° | 60° | 60° |
| LB$_{\lambda = 3.2}$ | −10° | −25° | −15° | −8° | −45 | −60° | −60° |
| UB$_{\lambda = 3.2}$ | 10° | 25° | 15° | 8° | 45° | 60° | 60° |

These bounds were chosen to allow the optimiser to explore as much as possible of the vaste parameter space within the mechanical limitations of the set-up to ensure safe operation of the turbine.

exceptionally high effective angles of attack on the turbine blade. Very high angles of attack are associated with highly inefficient individuals and excessive stress on the experimental apparatus. Similarly, increasing frequency terms in the Fourier series are given smaller amplitude constraints to avoid searching for power-hungry and high-torque pitch profiles. A strong indicator that our search space bounds is not over-constraining the optimisation procedure is that optimal individuals lie far from the boundaries.

The optimisation framework was fully automated to run unsupervised (Fig. 2). The genetic algorithm iteratively generated generations of 60 individuals, each of which is specified by a vector $\vec{x} = [A_0, A_1, A_2, A_3, \theta_1, \theta_2, \theta_3]^T$. The population size was selected as a good compromise between population diversity and convergence speed. Obtaining the fitness of an individual requires approximately 60 s. For a population size of 60 individuals, we can test a generation in an hour and achieve convergence in under two days of continuous operation.

The first generation consisted of randomly selected individuals assuming uniform distributions for each parameter constrained by the lower and upper bounds indicated in Table 1. Each individual was tested experimentally and their fitness was evaluated by computing both objectives functions (equation (6)). Once all individuals in the first generation have been evaluated, they were ranked according to their combined score across both objective functions. The ranking of individuals is based on their dominance relative to other individuals. An individual $\vec{p}$ dominates an individual $\vec{q}$ for a vector-valued objective function $\vec{f}$ if and only if $f_i(\vec{p}) \leq f_i(\vec{q})$ for all $i$ and $f_i(\vec{p}) < f_j(\vec{q})$ for some $j$[47]. Individuals that are not dominated by any other individuals are rank 1 and are Pareto-optimal. Rank 2 individuals are dominated only by rank 1 individuals and rank $n$ individuals are only dominated by rank $n-1$ individuals. All individuals of the same rank are considered equally good. Ranking is the key mechanism by which genetic algorithms promote the replication and propagation of elite individuals. The exact mechanism by which offsprings are generated from one generation to the next depends on the specific variant of genetic algorithm that is deployed.

The MATLAB toolbox deploys a controlled elitist genetic algorithm that is a variant of NSGA-II[46]. An elitist genetic algorithm promotes individuals with higher fitness rankings. Here, 5% of individuals for each new generation were clones of the previous generation's Pareto-optimal individuals. A controlled elitist genetic algorithm extends its preference to individuals who enhance population diversity, even if their fitness rankings are lower[43]. The procedure used to execute this selection is a binary tournament on the current population, whereby high ranked individuals inherently have a higher probability of being selected. The NSGA-II ensures population diversity in crossover individuals, which represent 60% of the offspring in our study. The crossover process creates children by taking a weighted average of the parents. The weighting is given by a random number between 0 and 1. Parents with a high rank and living in a sparsely populated region of the Pareto front have a higher likelihood of being selected for crossovers. Lastly, 35% of offspring are mutated individuals. A mutation is the random modification of one or several of the input parameters of a highly ranked individual, keeping the rest of the parameter untouched. The generation and evaluation of generations was iterated until the optimisation results converged.

There is no straight forward strategy to evaluate the convergence of a multi-objective genetic algorithm optimisation. Generally, the convergence of an optimisation problem is assessed from the statistical evolution of the cost-function. Pareto-based approaches minimise a vector of objective functions rather than a single cost function. A common metric to evaluate the convergence of Pareto fronts is the generational distance (GD), defined by[61] as the mean Euclidean distance between the new Pareto-optimal members $S$ and those from the previous generation $P$:

$$GD(S, P) = \frac{1}{|S|} \sqrt{\sum_{\vec{s} \in S} \min_{\vec{r} \in P} \| \vec{f}(\vec{s}) - \vec{f}(\vec{r}) \|}, \quad (8)$$

where $\vec{f}$ is the vector-valued objective function. For each Pareto-optimal member, the Euclidean distance is measured from the closest member in the previous Pareto Front. The objective functions were normalised by their mean value across all individuals from the first 20 generations to avoid introducing a bias in the convergence of either objective. The optimisation is considered converged when the generational distance and its gradient are $\leq 10^{-2}$ $1e$ for at least two consecutive generations. The convergence of the generational distance and its gradient is presented in Supplementary Fig. 4. This criterion was reached within 17 and 27 generations for tip-speed ratios $\lambda = 1.5$ and $\lambda = 3.2$, respectively. Both optimisations were allowed to run longer to monitor further evolution of the Pareto fronts.

**Particle image velocimetry**

Time-resolved planar particle image velocimetry (2D2C PIV) was used to measure the flow field around the wind turbine blade for selected individuals along the Pareto-front. A dual oscillator diode pumped ND:YLF laser (wavelength $\lambda = 527$ nm) with a maximum pulse energy of 30 mJ and a beam splitter were used to create two laser sheets from opposite sides of the water channel. The light sheets were oriented horizontally at mid-span of the turbine blade (Fig. 2). A high speed camera with a sensor size of 1024 px × 1024 px (Photron Fastcam SA-X2) and a spinning mirror apparatus were installed below the channel to capture the flow around the blade.

The spinning mirror apparatus consisted of two rotating and one stationary mirror, all oriented onto a 45° plane with respected to the horizontal plane. The two moving mirrors rotated about the same axis of rotation, at the same frequency as the wind turbine model. The outer mirror was placed at the same radius as the model blade, such that the blade was kept in the center of the field of view of the camera during the entire rotation. The field of view was 2.5 c × 2.5 c centred around the blade. The acquisition frequency was $f_{PIV} = 1000$ Hz. The images were processed using the commercial software PIVview (version 3.6.23 PIVtec GmbH/ILA_5150 GmbH) following standard procedures using a multi grid algorithm[62]. The final window size was 48 px × 48 px with an overlap of 75%. This yielded a grid spacing or physical resolution of 1.7 mm = 0.029c.

The uncertainty in the instantaneous velocity measurements using PIV is estimated by $\epsilon_{vel} = \frac{\epsilon_x}{\Delta t M}$, with $\epsilon_x$ the single displacement error in pixel, $\Delta t$ the time interval between the snapshots correlated by the PIV algorithm, and $M$ the magnification factor[62]. The single displacement error $\epsilon_x$ is the combination of the random error or measurement uncertainty $\epsilon_{rms}$ and the bias error $\epsilon_{bias}$. The random error is affected by the particle image diameter, the flow conditions, and the interrogation window size. For increasing interrogation window size, the measurement uncertainty decreases, but so also the spatial resolution decreases. The selected final window size of 48 px × 48 px gave us the best compromise between a high spatial resolution and a low random error. Based on careful analysis of the results, the random error was estimated smaller than 0.05 px for observation areas containing uniform flow and 0.1 px for observation areas where strong velocity gradients are present. The bias error is strongly affected by peak-locking, a phenomenon describing the tendency of the displacements to be biased towards integer pixel values. Histograms of subpixel displacement showed that peak locking was successfully avoided and the remaining bias error was assumed to be significantly less than the random noise error, i.e. $\epsilon_{bias} < 0.05$ px. The time interval between the snapshots was adapted as a function of the rotational phase as the effective velocity seen by the blade varies significantly

during the rotation (Fig. 1d). Velocity fields were obtained by correlating image number $n$ and $n + dn$, where $dn$ was determined such that the expected displacement at the instantaneous blade velocity would be above 12 px to reduce the relative error. The resulting relative PIV uncertainty is thus estimated to be $\epsilon_{vel}/U_{eff} < 0.0125$.

The maximum recording time on our camera was approximately 22 s, resulting in 21839 image pairs captured over 19 turbine rotations. The data is phase-averaged for 200 equally spaced bins such that 109 image pairs are averaged at each phase. The number of bins was selected as a compromise between obtaining satisfactory temporal discretisation of our phase-averaged results and a well converged mean flow. Based on the normalised root mean square error of the flow velocity magnitude, the phase averages are deemed sufficiently converged for all phases after 15 cycles for $\lambda = 1.5$ and after 10 cycles for $\lambda = 3.2$ (Supplementary Fig. 5).

### Reporting summary
Further information on research design is available in the Nature Portfolio Reporting Summary linked to this article.

## Data availability
The data sets generated during and/or analysed during the current study are publicly available on Zenodo https://doi.org/10.5281/zenodo.10776724.

## Code availability
All the relevant code used to generate the results in this paper are available from the corresponding author on request.

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

## Acknowledgements

This work was supported by the Swiss national science foundation under grant number PYAPP2_173652.

## Author contributions

Sébastien Le Fouest: conceptualisation, formal analysis, investigation, writing - original draft, visualisation; Karen Mulleners: conceptualisation, writing - review & editing, supervision, funding acquisition;

## Competing interests

All authors declare no competing interests.
