## [Peer Review File · Nature Communications]

REVIEWER COMMENTS

Reviewer #1 (Remarks to the Author):

Dear authors,

Please find here the details of my review of your work: Optimal blade pitch control for enhanced vertical-axis wind turbine performance

General comments

The present paper proposes an experimental implementation of optimization techniques based on genetic algorithms for the control of dynamic pitching for a horizontal wind turbine. The scientific theme, which is transdisciplinary, is well suited for Nature Communications and its topics. The paper is overall well written but is sometimes confusing as the objectives and details of the methodology are not fully explained. The experimental set-up, especially with the rotating mirrors device, is impressive and constitutes a significant asset for the research team. The machine learning technique chosen, the genetic algorithm, is now outdated in the flow control community, and the parameters of the algorithm are not justified in the text. I could accept the present work to be published in Nature Communications because of the experimental set-up and the significant results that could be obtained from this experiment. However, the current paper does not achieve a physical analysis of the results and remains at a qualitative level. In comparison, for comparable flow control studies using machine learning, publications which do not propose a full characterization of the optima are usually rejected from major journals. The bibliography must be completed with more references about machine learning for flow control and the physical phenomena of the wake. Major changes and reorganization of the paper are, in my opinion, needed to be able to fully understand the optimal solutions found by the algorithm.

Because of the 5000 words limit, I suggest removing one of the two configurations considered in the study, in order to fully deepen the other by more results. If minimal precisions and scientific rigor are not possible in the words limit, I would suggest another journal without this limit.

Specifics comments

Abstract

C1: It should be quantified in the abstract the main benefits (in %) of your method for all the configurations tested.

C2: Line 20: A reduction of 70% is cited for the off-design conditions, but this value cannot be found in the corresponding section. Please complete the off-design conditions paragraph or check this typo.

Literature review

C3: I think the literature review section should be rewritten and completed with more references. The three first paragraphs consist of well-known generalities on vertical wind turbines which do not correspond to our experimental one-blade wind turbine, and therefore simplified into few lines. However, there are significant missing elements including: (i) the physics of the flow dynamics under separation, reattachment and dynamical pitching; (ii) the use of genetic algorithms for flow control issues – please look for example at all the books of Steve Brunton’s team and colleagues on that subject (Brunton and Kutz (2022), Duriez et al. (2017)) – and especially for the trailing-edge/flow separation control on an airfoil that has been extensively studied in the past.

C4: Lines 59 and 60: you talked about “large-scale vortices on the surface of the turbine blades” without precisising the nature of them and for which physical phenomenon they correspond. Is it vortex shedding? boundary layer separation? Kelvin-Helmholtz instabilities? Please be more precise.

C5: Paragraph between lines 58 to 63: completing comment C4, an explanation of the physical phenomena existing in our configuration and interacting with our unsteady pitching is mandatory for the present paper.

C6: Line 65: control of a wind turbine blade separation using “plasma actuators” has been done in the past (Leroy et. al (2017), Zare Chavoshi et. al (2022)) and references must be added just after these words.

C7: Lines 67-72: these lines mix global optimization of the wind turbine operation and local optimization at the blade scale, which seems too confusing. I suggest you could separate the two and precise on what physical scales / frequencies of the flow these optimizations act.

C8: Line 80 / Equation 1: not necessarily at this position in the text, but you should explain how you obtain the power generated by the turbine from your measurements of the forces and moment on your experiment. I suppose you estimate the power from the moment measured ($P = M \omega$?), but it should be precise. It is also important as it is part of the objective function you optimized with your GA. I will also detail this later, but it is disappointing to see that you didn’t look at all on the effect of your pitching on the drag force measured in your experiment.

C9: Between 90 and 91: I would have expected here a full paragraph on the use of genetic algorithms for flow control configurations. A justification of the choice of this technique is here necessary, at this method is now outdated in the flow control community, who prefers using unsteady feedback control (genetic programming, deep reinforcement learning, etc.) which have been proved to perform much more efficiently for unsteady turbulent wakes. Please look at Brunton and Kutz (2022) for a good book of these techniques.

C10: Figure 1 (legend): even if I understood that the legends are not counted as words in the manuscript, I still think these very long legends in the figures do not help the full understanding of the figures. I think a simplification is required. Part of the legend (for Fig. 1 for example on the geometrical estimation of the effective angle and velocity) should be in the methods appendix. Significant information, like the definition of the TSR, should be in the main text.

C11: Paragraph between lines 91 – 105: references and numbers for quantification are needed here to justify the choice of your scenarios and especially if they correspond to real full-scale VAWT operations conditions.

C12: Page 6 / Equation 2: please justify why the frequency of pitching is limited to harmonics of the rotation frequency. This is linked to the previous comment C4 on the characteristic frequencies of the flow dynamics, and how your actuation frequencies you choose interact with them.

C13: Figure 2: Idem for the legend, much part of this correspond to the experimental set-up presentation and should be moved to the appendix.

Optimization of the pitching kinematics for off-design operation

C14: Paragraph from lines 113 to 123: I am not sure about the utility of this paragraph, especially when you are limited in words. A quantification with numbers is also necessary for this paragraph: for a full-scale VAWT, for what free-stream velocity the TSR is fixed? What about the complete deactivation of the wind turbine for very high wind speed (what is the TSR for that)? It is important to justify the choice of your TSR for your experiment, and if it corresponds to a proper scaling of the turbine.

C15: Line 124 and after: from this part of the text, you talk about the “bi-objective optimization” issue without clearly define the objectives. Usually for model-free data-driven techniques like genetic algorithms, we clearly define the equation of the cost function J that is used for the convergence of the algorithm (Brunton and Kutz (2022), Duriez et al. (2017)). Please write explicitly an equation for the cost function. You also have on this comment to be careful on two points: (i) as two different objectives are investigated here (the mean power coefficient and the fluctuations of the pitching moment), the ponderation between the two objectives must appears clearly in the cost function equation; (ii) considering comment C8, if the measured pitching moment is used to calculate the power coefficient as I assume, you have to precise if these objectives are really independent of each other. If they are not independent, it could significantly impact the convergence process of the genetic algorithm.

C16: Figure 3: as from the previous works on that subject (Brunton and Kutz (2022), Duriez et al. (2017)), detailing the parameters of all the individuals during the learning process is not useful if it is not coupled with a deep analysis of the physical reason of WHY the algorithm choose these individuals as optima. Consequently, two suggestions: or (i) only present the initial and the final generations, if you don't want to precisely explain the path followed by the algorithm; or, and it is my preference, (ii) explicit the evolution of the optimal individuals (with specific colors in Fig. 3(a)). It could be done for example by only presenting the best individuals of each generation (the ones selected for elitism), and for these

individuals, focus on the evolution of the main parameters (angle, C_p , M_z) and provide physical explanation.

C17: also Figure 3: please justify why there is no discussion on the other modes A1 and A2. For flow control applications, for bluff bodies in particular, low-frequency actuation is sometimes enough to improve significantly the aerodynamical performances of the set-up. As proposed in the general comments, focusing on only one configuration could help you pushing further the analysis of the GA results.

C18: Paragraph from lines 140 to 153: this analyze is mostly speculative and should be improved by the analysis of A1 and A2, and quantitative results from the best individuals (including spectral analysis).

C19: Figure 4 (legend): “exemplary optimal kinematics”: please precise in the legend and in the text (line 168) which case you considered here.

C20: also Figure 4: for the angles of $\theta = 45^\circ$ and 225° , where the C_p is minimum, is it true to say that the value of C_p cannot be changed because the pitching angle is never negative? if it seems to be due to limitations of the experimental set-up, it could be evoked in the perspectives for the set-up improvement.

C21: also Figure 4: Same question as C21, but with the dissymmetry between around $\theta = 135^\circ$ and 315° .

C22: Paragraph from lines 184 to 189: the interaction between the vortex shedding and the actuation performed seems also speculative and needs to be confirmed by (i) the characterization of the vortex shedding without pitching and (ii) spectral analysis of the optimal solutions.

C23: Figure 6: similarly for Fig. 3, for the increase of the second peak of C_p (by a factor 4 around $\theta = 315^\circ$), we also observe a significant increase of the pressure drag without significant change of the angle. However, the flow is fully attached for the non-actuated case. Please comment this in the text. As the drag is also measured in your experiment, I would analyze further the evolution of the drag for the optimal solutions found by GA.

C24: also Figure 6: Fig. 6(d) is difficult to read and could be simplified.

C25: Paragraph from lines 247 to 253: Same comment as C22

Discussions and perspectives

C26: based on my comments on the GA applications for flow configurations in the bibliography section, a perspective on the improvement of the machine learning technique used (with justification) would be valuable.

C27: based on my comment C20 on the limitations of the pitching angle range, a perspective on the improvement of the experimental set-up (with justification) would be valuable.

C28: lines 292-297: the remark that the turbulence has not impact any of the physical dynamics of the unsteady flow considered here is a very strong statement and needs to be supported by sources. Just by considering the instabilities present on the airfoil, I am not personally convinced by this statement. A quick look at the literature seems to disagree with that statement too (Kim and Xie (2016)).

C29: lines 298-303: a perspective on closed-loop control is interesting but should be much more precise. Precisions on the actuators is then needed, as their time of response needs to be quick, and all the actuators are not fitted for closed-loop control. According to my calculations, the vortex shedding is between 3 and 8 times higher than the pitching frequencies, the sheer layer instability between 4 and 13 times higher. An extensive study of the bibliography on this type of actuators and trailing edge actuation is therefore needed (Bak et al. (2007,2010), Carvalho et al. (2021)). You could also consider the same cost function as defined in the present study, to establish the link between this paper and the perspective.

Methods

Flow conditions

C30: lines 315-316: an acronym U_{∞} is necessary for the freestream velocity.

C31: line 327: a blockage ratio of 12.5% is defined as “relatively small”. For wind and water tunnels studies, it is still considered as high. Please provide a reference to compare this blockage ratio to other studies.

C32: line 331: A low-speed TSR value is considered for $\lambda < 4$, where in page 5 it is defined as $\lambda < 2.5$. What is the right value?

C33: line 336: “The rotating frequency is 0.89 Hz”: this is not a correct way to present a rotation speed, please favor rad/s or rpm. According to my calculations, a Reynolds number is obtained according to the equation of line 337 with ω in rad/s (here $\omega = 5.6$ rad/s).

C34: line 337: as this definition of the Reynolds number is independent of the freestream velocity, I suggest adding also the Reynolds based on the free-stream velocity. A proper discussion on the scaling between your model and a full-scale H-type VAWT would be also valuable, based on this Reynolds number especially.

C35: line 342: I am not sure the sentence starting as “The absolute position...” is necessary.

Forces measurements

C36: line 358: it seems that each phase of the phase-average force is based only on 25 cycles. Statistics on the convergence need to be precise in the methods section.

Genetic algorithms

C37: a scheme is needed to present the main steps of the genetic algorithm. Usually, previous works draw a scheme based on the one of Duriez et al. (2017).

C38: line 379: 60 individuals for each generation could be considered standard for experimental implementation of ML but needs to be justified based on the minimal time required for the estimation of each individual and the convergence. Please add a justification for this.

C39: Lines 385-394: the definition of the objectives needs to be explicated by the equation of the cost function. Please also refer to comment 15.

C40: Lines 396-397: "the best individuals were randomly mutated and recombined...": this sentence could be misleading because it suggests that mutation and crossover operations are reserved for the best individuals, which is not the case here. Traditionally, ALL the individuals will be submitted to genetic operations, except the best individuals which will go directly to the next generation by elitism. Please rephrase this section if this is how the algorithm works.

PIV

C41: The number of images pairs by phase needs to be precise in the text.

C42: Following C41, the statistics of the convergence for the PIV is needed, as well as the uncertainties of the PIV.

C43: line 422: Please indicate the software (with the version) used for the flow processing if an industrial code has been used. Otherwise please indicate "home-made software".

Technical corrections

Lines 35, 81, etc. You should uniformize the word "tip-speed-ratio" for which you use different spellings. The acronym TSR is widely used in the community, and you also define the Greek letter λ for that.

Line 222: the sentence needs to be rewritten.

Fig. 3 and 6: the notation for ω is maybe not uniformized (ω and Ω for the wind turbine rotation)

Lines 304-308: the sentence is difficult to understand and needs to be rephrased.

Line 232: "was kept constant at" -> is

Line 336: "was kept constant at" -> is

Line 368: "date" -> "data"

Line 398: "new new"

Line 398: "consited" -> "consisted"

Line 407: "PIV" -> "PIV2D2C"

Reviewer #2 (Remarks to the Author):

"Optimal blade pitch control for enhanced vertical-axis wind turbine performance"

Sébastien Le Fouest and Karen Mulleners

The authors describe a series of experiments, driven by an optimization algorithm, that explore the effects of active blade pitching on vertical-axis wind turbine (VAWT) performance. A space of seven waveform parameters is used for the optimization, and a large number of cases are considered for two candidate inflow conditions. The authors report significant power enhancements over the baseline (unactuated) case when the optimal pitch kinematics are used, while simultaneously reducing pitching-moment fluctuations. These results are compared with flow-field measurements to verify some hypotheses regarding the mechanisms responsible for these improvements.

I find that the paper is well-written, clear, and concise, and the results are of strong interest to the renewable-energy community. The main insight of the study is based in simple quasi-steady aerodynamics, i.e. that reducing the instantaneous angle of attack on the turbine blade decreases losses associated with flow separation and deep dynamic stall. This is perhaps not surprising from a theoretical standpoint, but the demonstration of the principle in VAWT applications is noteworthy. In this light, the relative simplicity and interpretability of the guiding narrative of the study is commendable.

In my opinion, the analysis in this study should be expanded upon so that the interesting results of the study are supported by a more robust discussion of the underlying flow physics. The theoretical foundations of the study could be established in more detail, so that the discussion of the results would be less hypothetical and more generalizable. I believe strengthening these elements of the manuscript would make the study more impactful and appropriate for publication in Nature Communications. Comments on these themes are given below, followed by a few minor suggestions.

Major comments:

1. I would prefer to see a stronger theoretical motivation for the choice and analysis of airfoil pitch kinematics, since the data-driven optimization approach is not particularly interpretable. For instance, the effects of the higher waveform harmonics are reported on, but no physical mechanisms for the observed effects are hypothesized or discussed. The trends in the results are mostly explained by means of the instantaneous effective angle of attack. If this is the case, however, then should not an analytical approach that seeks to minimize the effective angle of attack (as quantified by the velocity triangles shown in Figure 1) produce similar results? Such an analytical formulation would be more generalizable and informative for control and optimization purposes. The pitching moment could also be analytically tractable in attached-flow regimes, as even unsteady models such as that of Sears (1941) give predictions that could at least be used to guide or contextualize the data-driven approach. Such theoretical considerations would make it easier for the reader to grasp the consequences of these findings for turbines in real-world operating conditions.

2. The idea of controlling flow structures and loads in airfoils undergoing dynamic stall using active pitching is well-established in the literature, e.g. Prangemeier et al. (2010). I think it would help readers understand the particular novelty of this work if the authors were to provide more context, either via references to other studies or additional physics-based arguments, regarding the mechanisms responsible for the observed behaviors, such as the magnitude of the shed circulation and the direction of the vortex shedding.

3. The study does not account for power losses associated with actively pitching the turbine blade. The net power produced by the turbine will be the difference between the total power extracted and this active-pitch input power. If this consideration is added into the objective function for the optimization, are the same power enhancements possible?

4. The study also does not account for variations in the turbine rotation rate that would result as a function of the active-pitching control scheme. The turbine in this study is driven at a constant rotation rate, but for an operational VAWT with varying blade pitch angles, the torque on the rotor will vary as a function of the pitch waveform. These variations in torque will induce rotation-rate perturbations and thereby contribute to intracycle angle-of-attack variations. For a multi-bladed turbine with large rotor inertia, these effects may be reduced. Still, for the purposes of generalizing to real operating conditions, it would be beneficial if the authors could determine the extent to which intracycle variability in the rotation rate would change their findings (e.g. the magnitude of power enhancements).

Minor suggestions:

1. Induction effects are not considered in the angle-of-attack calculation. Could the authors comment on how their control schemes would need to be modified for a multi-bladed turbine with non-negligible induction factor?

2. Throughout, “upwind” and “downwind” are used as nouns; I would suggest that they should function as adjectives, i.e. “the upwind phase [of the rotation cycle].”

3. Lines 89-90: Perhaps “For these reasons, vertical-axis wind turbines typically operate at intermediate tip-speed ratios.”

4. The black arrows in Figures 4 and 6 cannot be differentiated from the black airfoil when the magnitude is small. Please consider using a different color.

References:

Prangemeier, T., D. Rival, and C. Tropea. “The Manipulation of Trailing-Edge Vortices for an Airfoil in Plunging Motion.” *Journal of Fluids and Structures* 26, no. 2 (February 1, 2010): 193–204. <https://doi.org/10.1016/j.jfluidstructs.2009.10.003>.

Sears, William R. “Some Aspects of Non-Stationary Airfoil Theory and Its Practical Application.” *Journal of the Aeronautical Sciences* 8, no. 3 (1941): 104–8. <https://doi.org/10.2514/8.10655>.

Reviewers' comments

Reviewer #1

General comments

The present paper proposes an experimental implementation of optimization techniques based on genetic algorithms for the control of dynamic pitching for a horizontal wind turbine. The scientific theme, which is transdisciplinary, is well suited for Nature Communications and its topics. The paper is overall well written but is sometimes confusing as the objectives and details of the methodology are not fully explained. The experimental set-up, especially with the rotating mirrors device, is impressive and constitutes a significant asset for the research team. The machine learning technique chosen, the genetic algorithm, is now outdated in the flow control community, and the parameters of the algorithm are not justified in the text. I could accept the present work to be published in Nature Communications because of the experimental set-up and the significant results that could be obtained from this experiment. However, the current paper does not achieve a physical analysis of the results and remains at a qualitative level. In comparison, for comparable flow control studies using machine learning, publications which do not propose a full characterization of the optima are usually rejected from major journals. The bibliography must be completed with more references about machine learning for flow control and the physical phenomena of the wake. Major changes and reorganization of the paper are, in my opinion, needed to be able to fully understand the optimal solutions found by the algorithm. Because of the 5000 words limit, I suggest removing one of the two configurations considered in the study, in order to fully deepen the other by more results. If minimal precisions and scientific rigor are not possible in the words limit, I would suggest another journal without this limit.

*We thank the reviewer for their time and their detailed questions and suggestions. We have addressed your comments and concerns in detail below. In response to your comments, we have made significant changes to the manuscript that are made clearly visible in the supplementary file **revisions.pdf** where new text passages are marked in blue and deleted text is ~~strikethrough~~.*

The associate editor has assured us that there is no word limit on the manuscript. This allowed us to provide more details on the methodology and the objectives. However, some of the additional information you requested has been provided in the Supplementary Information document to not increase the number of figures in the manuscript.

Specific comments

Abstract

1. It should be quantified in the abstract the main benefits (in %) of your method for all the configurations tested.

Thank you for your suggestion. We obtained two sets of Pareto optimal individuals for two wind conditions. For off-design wind conditions, Pareto optimal individuals achieve a power coefficient enhancement between 150 % and 215 % and a reduction of load fluctuations between 60 % and 77 %. For on-design wind conditions, the most power-enhancing individual achieves a power coefficient increase of 212 % and an increase in load fluctuations of 186 %, while the most load-reducing individual achieves a 29 % reduction in load fluctuations and a 7 % increase in power coefficient. We kept the most significant numbers achieved by our control strategy in the abstract. Both sets of Pareto optimal pitch profiles achieve a three-fold increase in power coefficient compared to the non-actuated turbine and, at off-design condition, blade pitching achieves a 77 % reduction in structure-threatening load fluctuations.

2. Line 20: A reduction of 70% is cited for the off-design conditions, but this value cannot be found in the corresponding section. Please complete the off-design conditions paragraph or check this typo.

We apologise for the confusion. We had given an intermediate value of the reduction in load fluctuations achieved by our set of Pareto-optimal pitch profiles. We replaced that value by our maximum reduction in load fluctuations (77 %) achieved with blade pitching, which we refer to in our analysis.

Literature review

3. I think the literature review section should be rewritten and completed with more references. The three first paragraphs consist of well-known generalities on vertical wind turbines which do not correspond to our experimental one-blade wind turbine, and therefore simplified into few lines. However, there are significant missing elements including: (i) the physics of the flow dynamics under separation, reattachment and dynamical pitching; (ii) the use of genetic algorithms for flow control issues – please look for example at all the books of Steve Brunton’s team and colleagues on that subject (Brunton and Kutz (2022), Duriez et al. (2017)) – and especially for the trailing-edge/flow separation control on an airfoil that has been extensively studied in the past.

We thank you for your suggestions. The literature review was thoroughly revised, and additional references were provided especially related to flow separation from pitching wings and the use of data driven methods for flow control.

4. Lines 59 and 60: you talked about “large-scale vortices on the surface of the turbine blades” without precisizing the nature of them and for which physical phenomenon they correspond. Is it vortex shedding? boundary layer separation? Kelvin-Helmholtz instabilities? Please be more precise.

The introduction and literature review has been thoroughly revised and now include a review of the stall development on unsteady airfoils.

5. Paragraph between lines 58 to 63: completing comment C4, an explanation of the physical phenomena existing in our configuration and interacting with our unsteady pitching is mandatory for the present paper.

We agree. Our revised introduction should give you a more comprehensive overview of the flow development on unsteady pitching airfoils.

6. Line 65: control of a wind turbine blade separation using “plasma actuators” has been done in the past (Leroy et. al (2017), Zare Chavoshi et. al (2022)) and references must be added just after these words.

Thank you for these references. We have added the reference Zare Chavoshi et. al (2022) in the manuscript and Aubrun et al. (2017), which we found even more relevant than Leroy et al. (2017) which seemed to be a conference paper.

7. Lines 67-72: these lines mix global optimization of the wind turbine operation and local optimization at the blade scale, which seems too confusing. I suggest you could separate the two and precise on what physical scales / frequencies of the flow these optimizations act.

Thank you for your suggestions which we adopted in our revised manuscript. The paragraph now also includes a discussion of the difference between the required actuation frequencies for control at the blade and the turbine scale.

8. Line 80 / Equation 1: not necessarily at this position in the text, but you should explain how you obtain the power generated by the turbine from your measurements of the forces and moment on your experiment. I suppose you estimate the power from the moment measured ($P = M\omega$?), but it should be precise. It is also important as it is part of the objective function you optimized with your GA. I will also detail this later, but it is disappointing to see that you didn't look at all on the effect of your pitching on the drag force measured in your experiment.

We apologise for the confusion. We do not obtain the extracted power from the measured pitching moment, but rather from the tangential force experienced by the turbine blade multiplied by the turbine's radius and rotational frequency. We then subtract the power required to perform the actuation to obtain a net generated power P_{net} :

$$P_{net}(\theta) = \underbrace{F_{\theta}(\theta)R\Omega}_{\substack{\text{power generated} \\ \text{by the blade}}} - \underbrace{M_Z(\theta)\dot{\alpha}(\theta)}_{\substack{\text{power consumed} \\ \text{by the actuation}}} \quad (1)$$

where F_{θ} and M_Z are the tangential force and pitching moment experienced by the turbine blade, and Ω and $\dot{\alpha}$ are the turbine's rotational frequency and the blade's pitch rate. This was previously explained in the methods section (lines 389 to 392), but we had not mentioned that the considered power is the net power production. This has been clarified in our revised manuscript and the equation for power production is now clearly highlighted in the methods section.

Regarding the drag, we are unsure whether you are referring to the negative tangential force acting on the turbine blade or the stream-wise force acting on the turbine's structure. We show and discuss the temporal development of the power coefficient, which is linearly related to the tangential force, so we indirectly discuss the drag. If you refer to the stream-wise force acting on the turbine's structure, that is another quantity that could indeed be evaluated and optimised for other scenarios than the ones we have focussed on in this manuscript. We have included this alternative optimisation as a perspective in the discussion section.

9. Between 90 and 91: I would have expected here a full paragraph on the use of genetic algorithms for flow control configurations. A justification of the choice of this technique is here necessary, at this method is now outdated in the flow control community, who prefers using unsteady feedback control (genetic programming, deep reinforcement learning, etc.) which have been proved to perform much more efficiently for unsteady turbulent wakes. Please look at

Brunton and Kutz (2022) for a good book of these techniques.

We thank you for your suggestion. Feedback control is accompanied by further challenges, including signal noise management and computational power. Given the inherent complexity of the flow around a vertical-axis wind turbines, we decided to perform open-loop control as a first step. Feedback control will be the subject of further investigation. Even though many novel data-driven methods for flow control have been proposed in recent year, they are typically demonstrated on numerical data or simple experiments. These new methods are not always well suited for noisy, experimental data. Our goal was not to test different methods, but focus on the potential of the blade pitching itself. Therefore, we build upon previous work where we used a genetic algorithm for the multi-objective optimisation of the pitching kinematics of flapping insect-inspired wings (Gehrke & Mulleners (2021)). Based on past experience, we know that genetic algorithms offer a robust and interpretable method for multi-objective optimisation of unsteady vortex dominated flows using complicated experimental set-ups. We added a full paragraph on the use of genetic algorithms for flow control configuration in the revised manuscript to offer context to our decision.

10. Figure 1 (legend): even if I understood that the legends are not counted as words in the manuscript, I still think these very long legends in the figures do not help the full understanding of the figures. I think a simplification is required. Part of the legend (for Fig. 1 for example on the geometrical estimation of the effective angle and velocity) should be in the methods appendix. Significant information, like the definition of the TSR, should be in the main text.

Thank you for the suggestion which we followed. The legends have been shortened and the information that was cut was moved to the main manuscript or the Supplementary Information document. This was easily possible as the associate editor informed us that there is no word limit for the manuscript.

11. Paragraph between lines 91 – 105: references and numbers for quantification are needed here to justify the choice of your scenarios and especially if they correspond to real full-scale VAWT operations conditions.

The scenarios we selected correspond indeed to real full-scale VAWT operating conditions. The exact values of the optimal tip-speed ratio and the corresponding off-design tip speed are dependent on the

specific geometry of the turbine. To justify our choice, we have included the experimental results of the power characterisation of our wind turbine model without actuation as a function of tip-speed ratio (Supplementary Fig. S1).

12. Page 6 / Equation 2: please justify why the frequency of pitching is limited to harmonics of the rotation frequency. This is linked to the previous comment C4 on the characteristic frequencies of the flow dynamics, and how your actuation frequencies you choose interact with them.

We use harmonics of the turbine frequency to enforce periodicity. Similar to what is done in Strom (2017), who performed intra-cycle control of the turbine rotational velocity, we used only the first three harmonics. The first three harmonics allow for sufficiently large local pitching gradients and phase shifts to change the extreme values of the effective angle of attack and when it exceeds its critical limits without introducing higher frequency vibrations. This justification has been added to the revised manuscript.

13. Figure 2: Idem for the legend, much part of this correspond to the experimental set-up presentation and should be moved to the appendix.

Thank you for the suggestion which we have followed for all figures.

Optimization of the pitching kinematics for off-design operation

14. Paragraph from lines 113 to 123: I am not sure about the utility of this paragraph, especially when you are limited in words. A quantification with numbers is also necessary for this paragraph: for a full-scale VAWT, for what free-stream velocity the TSR is fixed? What about the complete deactivation of the wind turbine for very high wind speed (what is the TSR for that)? It is important to justify the choice of your TSR for your experiment, and if it corresponds to a proper scaling of the turbine.

We agree with your comment and removed the paragraph. We refer to our response to your earlier comment #11 for the justification of our choice of the tip-speed ratio.

15. Line 124 and after: from this part of the text, you talk about the “bi-objective optimization” issue without clearly define the objectives. Usually for model-free data-driven techniques like genetic al-

gorithms, we clearly define the equation of the cost function J that is used for the convergence of the algorithm (Brunton and Kutz (2022), Duriez et al. (2017)). Please write explicitly an equation for the cost function. You also have on this comment to be careful on two points: (i) as two different objectives are investigated here (the mean power coefficient and the fluctuations of the pitching moment), the ponderation between the two objectives must appear clearly in the cost function equation; (ii) considering comment 8, if the measured pitching moment is used to calculate the power coefficient as I assume, you have to precise if these objectives are really independent of each other. If they are not independent, it could significantly impact the convergence process of the genetic algorithm.

We apologise for the lack of clarity in the description of our optimisation method. We are performing a multi-objective optimisation, based on two clearly defined optimisation objectives: increasing the power coefficient C_P and reducing the pitching moment standard deviation $\sigma(M_Z)$. Mathematically, our Pareto-based multi-objective optimisation can be defined as follows:

$$\text{minimise } \vec{f}(\vec{x}) := [C_P(\vec{x}), \sigma(M_Z(\vec{x}))], \quad (2)$$

where $\vec{x} = [A_0, A_1, A_2, A_3, \theta_1, \theta_2, \theta_3]^T$ is the parameter vector used to parametrise the blade pitching kinematics. We do not define a single cost-function to evaluate the performance of individuals. Instead, Pareto-based multi-objective optimisations rely on the concept of dominance. An individual \vec{p} dominates an individual \vec{q} for a vector-valued objective function \vec{f} if and only if $f_i(\vec{p}) \leq f_i(\vec{q})$ for all i and $f_j(\vec{p}) < f_j(\vec{q})$ for some j . Individuals that are not dominated by any other individuals are rank 1 and are Pareto-optimal. Rank 2 individuals are dominated only by rank 1 individuals and rank n individuals are only dominated by rank $n - 1$ individuals. All individuals of the same rank are considered equally good, so the algorithm does not favour one objective over another. Ranking is the key mechanism by which genetic algorithms promote the replication and propagation of elite individuals. This method avoids the need for a cost function and user-induced bias related to the weighting of objectives.

Our method was clarified in the revised manuscript. The absence of a cost function also entails there is no straightforward method to evaluate the convergence of Pareto-based optimisation methods. We included convergence estimations based on the generational distance, a common metric that quantifies the mean Euclidean distance between the new Pareto-optimal members and those from the previous generation. Regarding your comment (ii), both optimisation

objectives are independent as highlighted in our clarification in response to your comment #8.

16. Figure 3: as from the previous works on that subject (Brunton and Kutz (2022), Duriez et al. (2017)), detailing the parameters of all the individuals during the learning process is not useful if it is not coupled with a deep analysis of the physical reason of WHY the algorithm choose these individuals as optima. Consequently, two suggestions: or (i) only present the initial and the final generations, if you don't want to precisely explain the path followed by the algorithm; or, and it is my preference, (ii) explicit the evolution of the optimal individuals (with specific colors in Fig. 3(a)). It could be done for example by only presenting the best individuals of each generation (the ones selected for elitism), and for these individuals, focus on the evolution of the main parameters (angle, C_p , M_z) and provide physical explanation.

We thank you for your suggestion. We have included a figure showing the Pareto front evolution for subsequent generations in Supplementary Fig. S7. The algorithm converges quite fast towards the general shape of optimal individuals. Our generational distance convergence analysis shows a sub 5% change in Pareto-optimal scores after 5 to 7 generations (Supplementary Fig. S4). All generations past this state densify the Pareto front while achieving minor improvements through minor changes to blade kinematics. Note there is minimal change from one Pareto optimal set to the next past the 5th generation. Unfortunately, we were not able to derive any further insight from analysing the evolution of Pareto front in our case. Instead we focussed our analysis on the diversity within the Pareto optimal individuals identified by the multi-objective algorithm. In figures 4b and 6b, we present the evolution of the flow fields for a selected individual from each optimisation and modified figures 4c,d and 6c,d to summarise better the characteristic features that are shared between the Pareto optimal individuals.

17. also Figure 3: please justify why there is no discussion on the other modes A1 and A2. For flow control applications, for bluff bodies in particular, low-frequency actuation is sometimes enough to improve significantly the aerodynamical performances of the set-up. As proposed in the general comments, focusing on only one configuration could help you pushing further the analysis of the GA results.

Thank you for pointing this out. We show the coefficients A_0 and A_3 in the main manuscript as they show the clearest variations with

the optimisation objectives. We have now also included the figures for the other coefficients in Supplementary Fig. S2.

18. Paragraph from lines 140 to 153: this analyze is mostly speculative and should be improved by the analysis of A1 and A2, and quantitative results from the best individuals (including spectral analysis).

Here we have to apologise for not having taken any action. Unfortunately, we did not understand what you find speculative or what you have in mind when you suggest a spectral analysis. In this paragraph, we merely describe the overall pitching manoeuvres that are common for all Pareto front individuals. We think that this comment might be the result of the confusion we created by insufficiently describing the multi-objective optimisation and we hope that our improved description of the methods has also resolved this comment.

19. Figure 4 (legend): “exemplary optimal kinematics”: please precise in the legend and in the text (line 168) which case you considered here.

Thank you for pointing this out. We have added a symbol in figures 3a (respectively 5a) to indicate which individual we consider in figures 4b (6b).

20. also Figure 4: for the angles of $\theta = 45^\circ$ and 225° , where the C_p is minimum, is it true to say that the value of C_p cannot be changed because the pitching angle is never negative? if it seems to be due to limitations of the experimental set-up, it could be evoked in the perspectives for the set-up improvement.

The pitching angle is negative for all the Pareto optimal individuals during most of the upwind part of the cycle ($0^\circ < \theta < 180^\circ$) (figure 3b), which leads to a decrease in the effective angle of attack (figure 3c). The C_p is substantially varied by the blade pitching during the upwind for θ up to 180° which is clearly visible in the revised figure 4c where we show the angle at which C_p falls below 0. Blade pitching pushed that limit to $180^\circ < \theta < 190^\circ$. Further pushing that limit is not restricted by the experimental set-up but rather by the fact that the effective velocity is minimal at $\theta = 180^\circ$ (figure 1d).

We decided to constrain the A_1 term of our blade kinematics parametrisation to negative values to facilitate convergence and

avoid unnecessary stress on our wind turbine model. Our experimental setup offers the possibility to perform pitching profiles with a positive A_1 term. Preliminary experiments with positive A_1 resulted in extremely high angles of attack, which were detrimental to the wind turbine performance and we constrained the parameter space within reasonable boundaries to facilitate convergence. This has been specified in the revised manuscript. To ensure we did not over-constrain our parameter space, we show in Supplementary Fig. S6 that our optimal individuals lie far from the boundaries.

21. also Figure 4: Same question as 21, but with the dissymmetry between around $\theta = 135^\circ$ and 315° .

We assume that you are referring to the difference in the power coefficient peak for the actuated wind turbine blade between phases $\theta = 135^\circ$ and $\theta = 315^\circ$. This difference is not related to limitations in our experimental setup. Even if we would create a symmetric effective angle of attack profile, the evolution of the effective flow velocity is inherently asymmetric (figure 1). Furthermore, the blade interacts with clean incoming flow in the upwind, while the downwind involves blade-wake interactions even for a single-bladed wind turbine (Simão Ferreira et al. 2009) which adds to the asymmetry between up- and downwind. The differences observed are not related to any limitation of our experimental set-up.

22. Paragraph from lines 184 to 189: the interaction between the vortex shedding and the actuation performed seems also speculative and needs to be confirmed by (i) the characterization of the vortex shedding without pitching and (ii) spectral analysis of the optimal solutions.

We apologise for the unclarity and have improved figure 4c to better compare the timing of the vortex shedding or stall onset, which is accompanied by a drop in the power coefficient, between the non-actuated and the optimal solutions. For the non-actuated case, stall onset occurs at $\theta \approx 125^\circ$. For the optimal actuated cases, stall onset is delayed and occurs for $180^\circ < \theta < 190^\circ$. The rapid inward pitch manoeuvre occurring at the transition from upwind to downwind forces the stall vortex to shed and promotes flow reattachment at during the downwind phase. The potential to promote flow reattachment by a fast pitching manoeuvre has been demonstrated previously by Prangemeier et al. (2010) for plunging airfoils. This reference has also been added to the revised manuscript. We are intrigued by your suggestion for a spectral analysis, but we

are not sure what you have in mind. A Fourier analysis of the measured loads did not provide any arguments that we could exploit here. In our experience, most frequency analysis tools (Fourier analysis, DMD, etc.) do not provide much insight for highly transient flows. Relevant frequencies during the different stages of the flow development are hard to grasp as they occur only during a short window.

23. Figure 6: similarly for Fig. 3, for the increase of the second peak of C_p (by a factor 4 around $\theta = 315^\circ$), we also observe a significant increase of the pressure drag without significant change of the angle. However, the flow is fully attached for the non-actuated case. Please comment this in the text. As the drag is also measured in your experiment, I would analyze further the evolution of the drag for the optimal solutions found by GA.

We apologise, we are not sure what you mean by the drag force. We assume you are referring to the negative chord-wise force experienced by the turbine blade. This force is linearly proportional to the power coefficient, which we analyse in the manuscript.

We do understand the confusion that you highlight. The snapshots shown for $\theta = 315^\circ$ show very similar flow fields at similar angles of attack, yet the force experienced by the actuated blade is much higher than the force on the non-actuated blade. This is indeed an excellent example of the fact that the unsteady force response is strongly affected by the past evolution of the flow and is not solely governed by the instantaneous angles of attack and effective flow velocities. The accurate prediction of these unsteady history effects is still a major challenge for theoretical modelling but our data-driven approach is able to find dynamic pitch angle solutions that exploit them. We have revised our manuscript accordingly and we have also modified figure 4c to better support our discussion.

24. also Figure 6: Fig. 6(d) is difficult to read and could be simplified.

We apologise for the confusion. We have modified figures 4d and 6d to improve clarity and consistency.

25. Paragraph from lines 247 to 253: Same comment as 22

We realised that figure 6c was confusing as the turbine blade does not experienced stall at tip-speed ratio $\lambda = 3.2$. Therefore, we have replace figure 6c in the revised manuscript and updated its discussion as mentioned in our response to your comment #23.

Discussions and perspectives

26. based on my comments on the GA applications for flow configurations in the bibliography section, a perspective on the improvement of the machine learning technique used (with justification) would be valuable.

We thank you for your suggestion. Other optimisation methods, such as Bayesian optimisation, can also handle multi-objective optimisation and may have faster convergence times than our multi-objective genetic algorithm. As we managed to run one iteration per minute on our experimental setup, the convergence time was not our primary concern, but will be an important point of improvement for future studies. We believe that the multi-objective genetic algorithm did offer us a good compromise between convergence speed and robustness. We consider this method robust because it is unlikely to get stuck in a local minimum compared to gradient-based methods, especially in our non-linear problem, and it deals well with experimental noise. Genetic algorithm also offer some transparency and interpretability throughout the optimisation procedure, which is desirable when combined with an experimental setup. The results we obtained are highly satisfactory. The optimisation method we used was not a subject of research, but rather a tool we used. We consider our main perspectives of improvement to be: (i) moving on from open-loop to closed loop control, (ii) operating in more realistic and dynamic flow conditions, and (iii) conceiving a larger turbine with additional blades. Our optimisation method was a successful choice in our opinion and we do not wish to engage in a discussion on alternatives, particularly because we can not guarantee they would offer stronger results. We have recently experimented with reinforcement learning, but so far we have not been able to achieve the same performance enhancement on the turbine despite additional improvements to the data exchange pipeline.

27. based on my comment 20 on the limitations of the pitching angle range, a perspective on the improvement of the experimental set-up (with justification) would be valuable.

We hope that our responses to comments #20 and #21 have convinced you that our experimental set-up was not limited in the range of possible kinematics. Our parameter space boundaries were selected to guide the convergence of our optimisation framework based on preliminary experiments. Kinematics beyond those bounds were also tested and yielded inefficient turbine performance. The main limitations of our experimental setup are its dimensions that result in a low Reynolds number and non-negligible flow blockage,

as well as the fact that it is single-bladed. These limitations are addressed in the discussion.

28. lines 292-297: the remark that the turbulence has not impact any of the physical dynamics of the unsteady flow considered here is a very strong statement and needs to be supported by sources. Just by considering the instabilities present on the airfoil, I am not personally convinced by this statement. A quick look at the literature seems to disagree with that statement too (Kim and Xie (2016)).

We apologise for the brevity. Our statement was only concerned with the dynamic stall delay for pitching airfoils, which is found to be largely independent of the Reynolds number for clean inflow conditions. Based on the literature, it appears that the turbulent inflow conditions can have a larger influence than just the Reynolds number. More research is desirable that focusses on the influence of inflow turbulence on the performance of vertical-axis wind turbines. The paragraph has been rewritten accordingly.

29. lines 298-303: a perspective on closed-loop control is interesting but should be much more precise. Precisions on the actuators is then needed, as their time of response needs to be quick, and all the actuators are not fitted for closed-loop control. According to my calculations, the vortex shedding is between 3 and 8 times higher than the pitching frequencies, the sheer layer instability between 4 and 13 times higher. An extensive study of the bibliography on this type of actuators and trailing edge actuation is therefore needed (Bak et al. (2007,2010), Carvalho et al. (2021)). You could also consider the same cost function as defined in the present study, to establish the link between this paper and the perspective.

Following your suggestion in comment #7 we have now made a more clear distinction between control mechanisms that act at the blade scale and require higher actuation frequencies. In our future work, we plan to continue to focus on the blade pitching control, which acts at the turbine level and requires lower actuation frequencies. We have successfully demonstrated the potential of individual dynamic blade pitching but to increase the flexibility and adaptability of this control strategy for unsteady and turbulent inflow conditions, we need to develop a closed-loop control algorithm. We have rewritten the perspectives paragraph accordingly.

Methods - Flow conditions

30. lines 315-316: an acronym U_∞ is necessary for the freestream velocity.

The symbol has been added.

31. line 327: a blockage ratio of 12.5% is defined as “relatively small”. For wind and water tunnels studies, it is still considered as high. Please provide a reference to compare this blockage ratio to other studies.

We apologise for the confusion. A blockage ratio of 12.5% is indeed non-negligible. The flow obstruction leads to a power coefficient enhancement due to an acceleration of the incoming flow velocity, however, it will not substantially alter the timescales and the topology of the flow development stages. To mitigate the power coefficient enhancement related to flow obstruction, we focus our analysis on the power coefficient enhancement achieved by the actuation relative to the non-actuated case that is subjected to the same blockage. We have clarified this statement in the revised manuscript and added the reference to an investigation that analysed the influence of flow obstruction on a single-blade wind turbine up to a blockage ratio of 32%.

32. line 331: A low-speed TSR value is considered for $\lambda < 4$, where in page 5 it is defined as $\lambda < 2.5$. What is the right value?

We apologise for the typo, the value was corrected to $\lambda < 2.5$.

33. line 336: ”The rotating frequency is 0.89 Hz”: this is not a correct way to present a rotation speed, please favor rad/s or rpm. According to my calculations, a Reynolds number is obtained according to the equation of line 337 with ω in rad/s (here $\omega = 5.6$ rad/s).

This has been corrected. The rotational speed is $\Omega = 5.6 \text{ rad s}^{-1}$. We have used the capital Ω for the rotational speed in the revised manuscript to avoid confusion with the vorticity ω as you pointed out in the last comment #44.

34. line 337: as this definition of the Reynolds number is independent of the freestream velocity, I suggest adding also the Reynolds based on the free-stream velocity. A proper discussion on the scaling between your model and a full-scale H-type VAWT would be also valuable, based on this Reynolds number especially.

We thank you for your suggestion. Indeed, an alternative definition of the Reynolds number Re_D is obtained from the free-stream velocity and turbine diameter. This definition is generally relevant for farm-level investigations. We have included the values for Re_D for completeness in the revised document.

35. line 342: I am not sure the sentence starting as “The absolute position...” is necessary.

We have removed this sentence.

Methods - Forces measurements

36. line 358: it seems that each phase of the phase-average force is based only on 25 cycles. Statistics on the convergence need to be precise in the methods section.

We have added convergence plot in the Supplementary Fig. S3 for the two objective functions using force measurements for 100 cycles for a non-actuated turbine. The normalised root mean square error for the mean power coefficient and the standard deviation of the pitching moment drop below 1% after 25 cycles. These statistics are expected to improve with actuations given that they experience lighter flow separation and fewer cycle-to-cycle variations in the aerodynamic forces. This information has been added to the revised manuscript.

Methods - Genetic algorithms

37. a scheme is needed to present the main steps of the genetic algorithm. Usually, previous works draw a scheme based on the one of Duriez et al. (2017).

We thank you for your suggestion. We were not sure the schematic we found in Duriez et al. (2017) would be sufficient to describe the main steps of the algorithm. Therefore, we decided not to add the schematic, but to rewrite and expand instead the subsection on the genetic algorithm in the Methods section as that there is no longer a word limit. The revised description should address all comments in the section (from #37 to #40).

38. line 379: 60 individuals for each generation could be considered standard for experimental implementation of ML but needs to be justified based on the minimal time required for the estimation of each individual and the convergence. Please add a justification for this.

We thank you for your suggestion. A justification was added to the manuscript.

39. Lines 385-394: the definition of the objectives needs to be explicated by the equation of the cost function. Please also refer to comment 15.

As mentioned in our response to comment #15, there is not a single cost function in Pareto-based multi-objective genetic algorithms. Rather, the objective functions are minimised equally and in parallel creating a set of equally ranked Pareto-optimal individuals. The ranking procedure and convergence of our optimisation were clarified in the revised manuscript.

40. Lines 396-397: “the best individuals were randomly mutated and recombined...”: this sentence could be misleading because it suggests that mutation and crossover operations are reserved for the best individuals, which is not the case here. Traditionally, ALL the individuals will be submitted to genetic operations, except the best individuals which will go directly to the next generation by elitism. Please rephrase this section if this is how the algorithm works.

We apologise for the confusion. At each iteration, all the individuals are ranked based on their fitness defined by our two objective functions. In the selection procedure, the highest ranked individuals of the current population have a higher probability to be selected for genetic operations. We have rewritten and expanded the subsection on the Genetic Algorithm in the Methods section to address this comment and the previous ones related to the algorithm.

Methods - PIV

41. The number of images pairs by phase needs to be precise in the text.

The maximum recording time on our camera was approximately 22s, resulting in a total of 21 839 image pairs captured over 19 turbine rotations. The data is phase-averaged into 200 equally spaced bins such that 109 image pairs are averaged at each phase. This information has been added in the revised Methods section.

42. Following 41, the statistics of the convergence for the PIV is needed, as well as the uncertainties of the PIV.

Thank you for your reminder. We have estimated the PIV uncertainty and computed the normalised root mean square error (NRMSE) of the velocity magnitude for all phases to assess the convergence of the phase averaging procedure. As expected, the phase averages converge earlier for $\lambda = 3.2$ and for the phases at which the flow is attached at $\lambda = 1.5$. Overall, the phase averages are deemed sufficiently converged for all phases after 15 cycles for $\lambda = 1.5$ and after 10 cycles for $\lambda = 3.2$ (Supplementary Fig. S5). The revised manuscript now includes the estimation of the PIV uncertainty and convergence.

43. line 422: Please indicate the software (with the version) used for the flow processing if an industrial code has been used. Otherwise please indicate “home-made software”.

We have used the commercial software PIVview (version 3.6.23, PIVtec GmbH/ILA_5150 GmbH) This information has been added in the revised Methods section.

Technical corrections

44.
 - Lines 35, 81, etc. You should uniformize the word “tip-speed-ratio” for which you use different spellings. The acronym TSR is widely used in the community, and you also define the Greek letter λ for that.
 - Line 222: the sentence needs to be rewritten.
 - Line 222: the sentence needs to be rewritten.
 - Fig. 3 and 6: the notation for ω is maybe not uniformized (ω and ω for the wind turbine rotation)
 - Lines 304-308: the sentence is difficult to understand and needs to be rephrased.
 - Line 232: “was kept constant at” → is
 - Line 336: “was kept constant at” → is
 - Line 368: “date” → “data”
 - Line 398: “new new”
 - Line 398: “consited” → “consisted”
 - Line 407: “PIV” → “PIV2D2C”

Thank you for pointing these out, they have been corrected.

Reviewer #2

The authors describe a series of experiments, driven by an optimization algorithm, that explore the effects of active blade pitching on vertical-axis wind turbine (VAWT) performance. A space of seven waveform parameters is used for the optimization, and a large number of cases are considered for two candidate inflow conditions. The authors report significant power enhancements over the baseline (unactuated) case when the optimal pitch kinematics are used, while simultaneously reducing pitching-moment fluctuations. These results are compared with flow-field measurements to verify some hypotheses regarding the mechanisms responsible for these improvements. I find that the paper is well-written, clear, and concise, and the results are of strong interest to the renewable-energy community. The main insight of the study is based in simple quasi-steady aerodynamics, i.e. that reducing the instantaneous angle of attack on the turbine blade decreases losses associated with flow separation and deep dynamic stall. This is perhaps not surprising from a theoretical standpoint, but the demonstration of the principle in VAWT applications is noteworthy. In this light, the relative simplicity and interpretability of the guiding narrative of the study is commendable. In my opinion, the analysis in this study should be expanded upon so that the interesting results of the study are supported by a more robust discussion of the underlying flow physics. The theoretical foundations of the study could be established in more detail, so that the discussion of the results would be less hypothetical and more generalizable. I believe strengthening these elements of the manuscript would make the study more impactful and appropriate for publication in Nature Communications. Comments on these themes are given below, followed by a few minor suggestions.

*We thank the reviewer for their time and their comments and suggestions. We have addressed your comments in detail below. In response to your comments, we have made significant changes to the manuscript that are made clearly visible in the supplementary file **revisions.pdf** where new text passages are marked in **blue** and deleted text is **strikethrough**.*

Major comments

1. I would prefer to see a stronger theoretical motivation for the choice and analysis of airfoil pitch kinematics, since the data-driven optimization approach is not particularly interpretable. For instance, the effects of the higher waveform harmonics are reported on, but no physical mechanisms for the observed effects are hypothesized or discussed. The trends in the results are mostly explained by means of the instantaneous effective angle of attack. If this is the

case, however, then should not an analytical approach that seeks to minimize the effective angle of attack (as quantified by the velocity triangles shown in Figure 1) produce similar results? Such an analytical formulation would be more generalizable and informative for control and optimization purposes. The pitching moment could also be analytically tractable in attached-flow regimes, as even unsteady models such as that of Sears (1941) give predictions that could at least be used to guide or contextualize the data-driven approach. Such theoretical considerations would make it easier for the reader to grasp the consequences of these findings for turbines in real-world operating conditions.

Thank you for your suggestion. We have thoroughly revised and expanded our introduction in response to the reviewers' comments including the motivation and goal of individual blade pitching. Blade pitching is a control strategy that acts at the turbine level to control the overall performance of vertical-axis turbines. The required actuation frequencies are lower than for control strategies at the blade scale, such as surface actuators. Surface actuators target the fast evolution of the blade's boundary layer whereas the turbine control measures target the evolution of the effective velocity and angle of attack within a turbine rotation to change the time at which critical conditions would be reached.

Even though the effectiveness of the optimal pitching kinematics can be explained relatively simple, the flow and force response to the unsteady conditions is non-linear and quasi-steady aerodynamic models fail in predicting them. The aerodynamics of a vertical-axis wind turbine blade are rather complicated, as they result from the combination of pitching and surging along a curved path. Analytical models such as those by Theodorsen (1935) and Sears (1941) look at the combined effect of pitching and surging, but they assume a straight wake and fail when flow separation occurs, which is the case even for the on-design operation. We are not aware of any dynamic stall models that extend to wings operating along a circular path neither.

We identified the effective angle of attack as is the most intuitive parameter to explain the performance enhancement achieved by our control framework. Our analysis is focussed on extracting common features that are present among the Pareto-optimal solutions that allow us to understand how the actuation manipulates the timescales and flow structures to improve power performance. However, the step from explaining the behaviour in hindsight to predicting it is still large. Minimizing the effective angle of attack would reduce the force on the blade and would be counterproductive. Our straight-bladed H-type wind turbine is lift-based and require a positive effec-

tive angle of attack variation to produce torque. Our optimisation seeks pitch profiles that allow the formation of flow structure at intermediate angle of attack for power production, and controlling the timing of their shedding to avoid detrimental consequences related to the separation of these flow structures. Given the complexity of the flow we are investigating, a data-driven approach is currently the most promising strategy to find optimal blade pitching kinematics.

2. The idea of controlling flow structures and loads in airfoils undergoing dynamic stall using active pitching is well-established in the literature, e.g. Prangemeier et al. (2010). I think it would help readers understand the particular novelty of this work if the authors were to provide more context, either via references to other studies or additional physics-based arguments, regarding the mechanisms responsible for the observed behaviors, such as the magnitude of the shed circulation and the direction of the vortex shedding.

Thank you for bringing this reference to our attention. Even though they focus on the trailing edge vortex, they also observe the effect of the fast pitching motion on promoting flow reattachment. We have included the reference in the revised manuscript. Due to the limited size of our PIV field of view, we were not able to extract the trajectory of the vortex after shedding. We are currently analysing wake velocity data downstream of the turbine that show the effect of the shed vortices on the turbine wake. However, including this data would go beyond the scope of the current manuscript.

3. The study does not account for power losses associated with actively pitching the turbine blade. The net power produced by the turbine will be the difference between the total power extracted and this active-pitch input power. If this consideration is added into the objective function for the optimization, are the same power enhancements possible?

We did account for the power losses associated with the pitching motion, and we apologise that this was not clear in the initial manuscript. We obtain the turbine's power production from the tangential force experienced by the turbine blade multiplied by the turbine's radius and rotational frequency. We then subtract the power required to perform the actuation to obtain a net generated power P_{net} :

$$P_{net}(\theta) = \underbrace{F_{\theta}(\theta)R\Omega}_{\text{blade-generated power}} - \underbrace{M_Z(\theta)\dot{\alpha}(\theta)}_{\text{actuation-consumed power}} \quad (3)$$

where F_{θ} and M_Z are the tangential force and pitching moment ex-

perceived by the turbine blade, and Ω and $\dot{\alpha}$ are the turbine's rotational frequency and the blade's pitch rate. These equations are now included in the manuscript which has been revised to improve clarity about the power calculations.

4. The study also does not account for variations in the turbine rotation rate that would result as a function of the active-pitching control scheme. The turbine in this study is driven at a constant rotation rate, but for an operational VAWT with varying blade pitch angles, the torque on the rotor will vary as a function of the pitch waveform. These variations in torque will induce rotation-rate perturbations and thereby contribute to intra-cycle angle-of-attack variations. For a multi-bladed turbine with large rotor inertia, these effects may be reduced. Still, for the purposes of generalizing to real operating conditions, it would be beneficial if the authors could determine the extent to which intracycle variability in the rotation rate would change their findings (e.g. the magnitude of power enhancements).

We thank you for pointing this out. Indeed, the moment imparted by the pitching motor to the turbine blade is transmitted to the central shaft, altering the moment experienced by the turbine's central shaft. Here, the pitching moment is two orders of magnitude smaller than the moment experienced by the central shaft on average, so this effect is not a concern. On a real wind turbine, the rotational frequency is controlled by the generator to maintain a constant value, given that the torque production achieved by the turbine blades is not intermittent. Any additional variation in the moment experienced by the central shaft related to the actuation would also be controlled by the generator. Additionally, we expect the moment related to the actuation to approximately cancel out on a real wind turbine with multiple blades, as the optimal pitching kinematics are fairly periodic and symmetric in amplitude. When one blade pitches up, another pitches down. Nevertheless, we agree that a single short and lightweight turbine blade is advantageous for active blade pitching. This advantage is a limitation of our experimental apparatus when generalising results to a large-scale wind turbine. We have highlighted this limitation in the revised manuscript.

Minor suggestions

5. Induction effects are not considered in the angle-of-attack calculation. Could the authors comment on how their control schemes would need to be modified for a multi-bladed turbine with non-negligible induction factor?

Indeed, induction effects are not considered in the angle of attack calculations. These are inherently accounted for by our experiment procedure. The effective angle of attack plots we present are given to interpret the mechanisms by which the optimal pitching kinematics achieve performance enhancement. We do not inform the control framework with the expected geometric angle of attack variation experienced by the wind turbine blade. A multi-bladed turbine would experience higher induction and additional blade-wake interactions. We do not expect our control scheme to be greatly affected by additional blades in the upwind, given that the blades interact with unperturbed flow and induction effect are small until the transition to downwind. The downwind half would be significantly affected by the additional blades. We would argue that a successful control scheme would focus on achieving power extraction upwind, given that the downwind will be dominated by blade-wake interactions and momentum deficit. Further work will involve closed-loop control that should seamlessly extend to multiple blades. A small note in this regard has been added in the perspectives.

6. Throughout, “upwind” and “downwind” are used as nouns; I would suggest that they should function as adjectives, i.e. “the upwind phase [of the rotation cycle].”

We have modified the text as per your suggestion.

7. Lines 89-90: Perhaps “For these reasons, vertical-axis wind turbines typically operate at intermediate tip-speed ratios.”

We have modified the text as per your suggestion.

8. The black arrows in Figures 4 and 6 cannot be differentiated from the black airfoil when the magnitude is small. Please consider using a different color.

The color of the arrows has been changed.

REVIEWERS' COMMENTS

Reviewer #1 (Remarks to the Author):

The line numbers indicated in the current review correspond to the revised manuscript with explicit corrections.

General comments

I thank the authors for the significant improvements and changes of the manuscript, and the full response of my numerous comments. I find the paper much more complete, in terms of validation of the experimental set-up, justifications and results explanations. I have still few unmet expectations, in particular the physical analysis of the optimal solutions obtained with the multi-objective genetic algorithms and the use of GA method itself, but could be considered as future works. Therefore, I agree for the publication in Nature Communications.

Specifics comments

C1: Lines 131 to 147: I thank you for the rewriting of this paragraph, detailing the choice of the genetic algorithms, and I agree with this paragraph. I however disagree on your response of comment C9 where you described genetic algorithms as a “robust [...] method”: this is usually considered as optimal optimization and not robust when the initial conditions change.

C2: Lines 95-97: I don't understand the sentence “Their required actuation frequencies are lower than for the surface actuators as turbine control measures target the evolution of the effective velocity and angle of attack within a turbine rotation to affect the stall onset and vortex shedding.” Please rephrase. The use of “surface actuators” could be confusing, as you quote in lines 83-86 plasma and blowing jets corresponding “boundary layer control” or “separation control” actuators, with significant differences in terms of functioning. Pulsed jets characteristic frequencies are of the order of $O(100\text{Hz})$, where plasma actuators can reach $O(1\text{kHz})$.

C3: (response to comment 22): thank you for your precisions. Concerning the spectral analysis, I still think that it is an important tool to describe the unsteady flows. For transient flows, you could have extracted phase-locked fluctuating velocities from the PIV fields based on the angles, and compute statistics (RMS, spectrum) for each phase. The difficulty would be to choose a significant duration for the phase, corresponding to the actuation time scale, but may be limited due to the number of cycles accessible with the current database. A longer database could work.

C3: (response to comment 23): I apologize for the confusion with the drag, as the way to obtain the power coefficient was not clearly precised. I understood your comment, but if you indicate that the similar flow fields topologies give very different on the forces, it means the description of the results with the phase-averaged velocity may not be sufficient for the physical explanation of the controlled flow.

C4: (response to comment 26): same comment of C1 concerning the GA described as “robust”. Describing the results obtained from GA as “satisfactory” needs to be quantify with proofs and numbers (like the convergence and quantification I asked in my previous comments, and which is more detailed now). For example, we can argue that your optima found for this configuration would not be valid anymore for another Reynolds number or with incoming perturbations / turbulence, which is usually the case for optimal control.

C5: (response to comment 37): It is fine for me, the schematic is not mandatory and an explanation is enough.

Technical corrections

Line 21: Space missing before the “Based”

Line 74: leading to ?

Line 349: “This is effect is clearly visible...” -> I don’t understand the beginning of this sentence, it could be a typo.

References: there are several references [14,44,46,62] with issues (“???”)

Reviewer #2 (Remarks to the Author):

I appreciate the authors’ responses to my initial comments on their manuscript and their revisions to the manuscript draft. Even though I would still prefer to see a more centralized treatment of the flow physics being identified through the data-driven optimization scheme, I recognize that this is likely a matter of personal opinion and believe the paper presents sufficiently strong results to warrant publication. I suggest a physics-centered reframing of the story below in case it is helpful to the authors but leave to their discretion the extent to which they act on my feedback.

Main Suggestion

The additional discussion of the dynamics (lines 330-365 of the revised manuscript) underscores the point I had intended but failed to make in Comment 1 of my original review. When I suggested an

analytical approach to “minimize the effective angle of attack” (hereafter AoA), I intended to refer to minimizing the overshoot of the effective AoA above the static stall angle. (Clearly, as the authors pointed out, minimizing the instantaneous AoA itself would be counterproductive.) I apologize for this rather glaring miscommunication.

To my understanding, the authors make precisely the point I had intended to convey in lines 336-344, which is further borne out in Figures 3c and 5c – the data-driven optimization approach reduces the amplitude of the instantaneous effective AoA to near or below the static stall angle. My original comment was that this intuitive result could be arrived at qualitatively with a simple quasi-steady approximation of the aerodynamics in the blade frame, i.e. using the velocity triangles in Figure 1. This would give an educated guess for blade-pitch kinematics that could reduce the effective AoA, and the data-driven approach could then be employed from this starting point to capture the additional unsteady effects in the problem (referenced in lines 362-363*). In other words: start from a simplified physical model and use data-driven optimization to compensate for additional nonlinear effects. To some degree, the authors have already done this in the formulation of the optimization problem: in lines 576-584, they constrain the parameter space based on physical intuition for the effects of AoA on performance and unsteady loads. The study is thus already “physics-informed” in a sense.

My recommendation, therefore, is to present the study in the paper as being driven primarily by flow physics (i.e. AoA considerations) that are augmented by a data-driven optimization routine, and not the other way around. To me, the current presentation of the study centers on the data-driven method, with physical intuition being employed after the fact to qualitatively explain some of the results. This is what prompted my initial concerns with the theoretical contributions and generalizability of the work, as well as perhaps the comment of the other reviewer that “the current paper does not achieve a physical analysis of the results and remains at a qualitative level.” If the authors were to center the work around the physics-based hypothesis that dynamic pitching can avoid stall, rather than mention this as something that a data-driven method just happened to stumble upon over the course of a black-box exploration, readers would more readily grasp the motivating idea and the generalizability of their experimental findings. While this would require a reconfiguration of the manuscript itself, I don’t think it would necessitate additional experiments or analysis. It is my opinion that such a narrative shift would give the paper more credibility in the fluid-mechanics and aerodynamics communities, and would forestall (pun intended) criticisms from machine-learning audiences regarding the lack of novelty of the genetic-algorithm approach, since it would be understood as a modeling tool rather than the main contribution of the study.

*One final note: I tend to disagree with the author’s implicit assertion in lines 362-365 that, just because unsteady aerodynamic effects are hard to predict in theory, a fully data-driven approach is the only solution. While these effects may be hard to model in a quantitatively predictive sense, first-order approaches like quasi-steady models or the Sears and Theodorsen functions may give enough insight into the dynamics to get us started along the right path. Examples from the literature include the work of Baik et al. (2012), who showed that modifications to the Theodorsen function can produce qualitatively accurate representations of the forces on a plunging airfoil even in dynamic stall, and Brunton and Rowley (2013), who developed an empirically informed version of the Theodorsen function for closed-loop control on a pitching airfoil.

Minor Comments

Line 411: “Would have greater actuation costs”.

Line 427: I am not entirely sure that dynamic stall time delays are fully independent of Reynolds number, particularly for utility-scale turbine blades. See Kiefer et al. (2022), Section 7.

Line 505: Re_c is not constant in a VAWT, since it depends on the inflow velocity as well as the tangential velocity of the blade. I believe this should be presented as a range.

Line 583: “Fourier”.

References

Yeon Sik Baik, Luis P. Bernal, Kenneth Granlund, and Michael V. Ol. “Unsteady Force Generation and Vortex Dynamics of Pitching and Plunging Aerofoils.” *Journal of Fluid Mechanics* 709 (2012): 37–68. <https://doi.org/10.1017/jfm.2012.318>.

Steven L. Brunton and Clarence W. Rowley. “Empirical State-Space Representations for Theodorsen’s Lift Model.” *Journal of Fluids and Structures* 38 (2013): 174–86. <https://doi.org/10.1016/j.jfluidstructs.2012.10.005>.

Janik Kiefer, Claudia Brunner, Martin O. L. Hansen, and Marcus Hultmark. “Dynamic Stall at High Reynolds Numbers Induced by Ramp-Type Pitching Motions.” *Journal of Fluid Mechanics*. 938 (2022): A10. <https://doi.org/10.1017/jfm.2022.70>.

Reviewers' comments

Reviewer #1

General comments

I thank the authors for the significant improvements and changes of the manuscript, and the full response of my numerous comments. I find the paper much more complete, in terms of validation of the experimental set-up, justifications and results explanations. I have still few unmet expectations, in particular the physical analysis of the optimal solutions obtained with the multi-objective genetic algorithms and the use of GA method itself, but could be considered as future works. Therefore, I agree for the publication in Nature Communications.

Thank you very much for your positive evaluations and the valuable suggestions that help us improve the manuscript.

Specific comments

1. Lines 131 to 147: I thank you for the rewriting of this paragraph, detailing the choice of the genetic algorithms, and I agree with this paragraph. I however disagree on your response of comment C9 where you described genetic algorithms as a “robust [...] method”: this is usually considered as optimal optimization and not robust when the initial conditions change.

We indeed used robust to indicate that we obtained the same results for different initial condition and were not aware this is not the common interpretation. Thank you for pointing that out.

2. Lines 95-97: I don't understand the sentence “Their required actuation frequencies are lower than for the surface actuators as turbine control measures target the evolution of the effective velocity and angle of attack within a turbine rotation to affect the stall onset and vortex shedding.” Please rephrase. The use of “surface actuators” could be confusing, as you quote in lines 83-86 plasma and blowing jets corresponding “boundary layer control” or “separation control” actuators, with significant differences in terms of functioning. Pulsed jets characteristic frequencies are of the order of O(100Hz), where plasma actuators can reach O(1kHz).

*We apologise for the unclarity. The sentence previously on lines 95-97 has been rephrased.
Even though different surface actuators have different functioning*

mechanisms and operational frequencies, they all operate at frequencies that are one or more orders of magnitude above the turbine frequency, and therefore, we believe they can be grouped together.

3. (response to comment 22): thank you for your precisions. Concerning the spectral analysis, I still think that it is an important tool to describe the unsteady flows. For transient flows, you could have extracted phase-locked fluctuating velocities from the PIV fields based on the angles, and compute statistics (RMS, spectrum) for each phase. The difficulty would be to choose a significant duration for the phase, corresponding to the actuation time scale, but may be limited due to the number of cycles accessible with the current database. A longer database could work.

Thank you for clarifying. This could indeed be an interesting additional analysis, but it would require a much larger database. We will keep this in mind for future work.

4. (response to comment 23): I apologize for the confusion with the drag, as the way to obtain the power coefficient was not clearly precised. I understood your comment, but if you indicate that the similar flow fields topologies give very different on the forces, it means the description of the results with the phase- averaged velocity may not be sufficient for the physical explanation of the controlled flow.

We apologise for the confusion. Similar flow fields give different forces when the blades are at different angles of attack. The convergence study included in the supplementary material shows that the cycle-to-cycle repeatability of the flow response is high and the phase-averaged flow fields give a correct representation of the flow topology for the given kinematics.

5. (response to comment 26): same comment of C1 concerning the GA described as “robust”. Describing the results obtained from GA as “satisfactory” needs to be quantify with proofs and numbers (like the convergence and quantification I asked in my previous comments, and which is more detailed now). For example, we can argue that your optima found for this configuration would not be valid anymore for another Reynolds number or with incoming perturbations / turbulence, which is usually the case for optimal control.

The effect of incoming perturbations and turbulence is indeed of great interest and part of our ongoing work!

6. (response to comment 37): It is fine for me, the schematic is not mandatory and an explanation is enough.

Thank you.

Technical corrections

7.
 - Line 21: Space missing before the “Based”
 - Line 74: leading to ?
 - Line 349: “This is effect is clearly visible...” I don’t understand the beginning of this sentence, it could be a typo.
 - References: there are several references [14,44,46,62] with issues (“???”)

Thank you for pointing these out, they have been corrected.

Reviewer #2

I appreciate the authors' responses to my initial comments on their manuscript and their revisions to the manuscript draft. Even though I would still prefer to see a more centralized treatment of the flow physics being identified through the data-driven optimization scheme, I recognize that this is likely a matter of personal opinion and believe the paper presents sufficiently strong results to warrant publication. I suggest a physics-centered reframing of the story below in case it is helpful to the authors but leave to their discretion the extent to which they act on my feedback.

Thank you very much for your assessment and the valuable suggestions that help us improve the manuscript. Even though we appreciate your remaining suggestion to reframe the story, we stand behind the way we frame the story and have decided not to throw everything around at this stage.

Main suggestion

The additional discussion of the dynamics (lines 330-365 of the revised manuscript) underscores the point I had intended but failed to make in Comment 1 of my original review. When I suggested an analytical approach to “minimize the effective angle of attack” (hereafter AoA), I intended to refer to minimizing the overshoot of the effective AoA above the static stall angle. (Clearly, as the authors pointed out, minimizing the instantaneous AoA itself would be counterproductive.) I apologize for this rather glaring miscommunication. To my understanding, the authors make precisely the point I had intended to convey in lines 336-344, which is further borne out in Figures 3c and 5c - the data-driven optimization approach reduces the amplitude of the instantaneous effective AoA to near or below the static stall angle. My original comment was that this intuitive result could be arrived at qualitatively with a simple quasi-steady approximation of the aerodynamics in the blade frame, i.e. using the velocity triangles in Figure 1. This would give an educated guess for blade-pitch kinematics that could reduce the effective AoA, and the data-driven approach could then be employed from this starting point to capture the additional unsteady effects in the problem (referenced in lines 362-363*). In other words: start from a simplified physical model and use data-driven optimization to compensate for additional nonlinear effects. To some degree, the authors have already done this in the formulation of the optimization problem: in lines 576-584, they constrain the parameter space based on physical intuition for the effects of AoA on performance and unsteady loads. The study is thus already “physics-informed” in a sense. My recommendation, therefore, is to present the

study in the paper as being driven primarily by flow physics (i.e. AoA considerations) that are augmented by a data-driven optimization routine, and not the other way around. To me, the current presentation of the study centers on the data-driven method, with physical intuition being employed after the fact to qualitatively explain some of the results. This is what prompted my initial concerns with the theoretical contributions and generalizability of the work, as well as perhaps the comment of the other reviewer that “the current paper does not achieve a physical analysis of the results and remains at a qualitative level.” If the authors were to center the work around the physics-based hypothesis that dynamic pitching can avoid stall, rather than mention this as something that a data-driven method just happened to stumble upon over the course of a black-box exploration, readers would more readily grasp the motivating idea and the generalizability of their experimental findings. While this would require a reconfiguration of the manuscript itself, I don’t think it would necessitate additional experiments or analysis. It is my opinion that such a narrative shift would give the paper more credibility in the fluid-mechanics and aerodynamics communities, and would forestall (pun intended) criticisms from machine-learning audiences regarding the lack of novelty of the genetic-algorithm approach, since it would be understood as a modeling tool rather than the main contribution of the study.

We absolutely acknowledge that the solutions found by the algorithm make perfect sense. One of the main characteristics of the solution is to keep an effective angle close to the static stall angle. However, it remains non-trivial to decide when to increase the angle up to that level and when to switch from the positive to the negative effective angle values. Quasi-steady aerodynamic model typically do a decent job in predicting the overall trend of the forces, but typically fail at predicting the timing of the flow evolution that vortex dominated. This is not surprising as most of these models do not include the effect of leading edge separation or the formation of large scale vortices. Therefore, the approach you suggest is deemed to fail. However, in ongoing collaborative work with the University of Minnesota, we can show that the prediction of the aerodynamic loads based on sparse sensor data can be substantially improved when including the quasi-steady force predictions as a feature in the model.

*One final note: I tend to disagree with the author’s implicit assertion in lines 362-365 that, just because unsteady aerodynamic effects are hard to predict in theory, a fully data-driven approach is the only solution. While these effects may be hard to model in a quantitatively predictive sense, first-order approaches like quasi-steady models or the Sears and Theodorsen functions may give enough insight into the dynamics to get us started along the right path. Examples from the literature include

the work of Baik et al. (2012), who showed that modifications to the Theodorsen function can produce qualitatively accurate representations of the forces on a plunging airfoil even in dynamic stall, and Brunton and Rowley (2013), who developed an empirically informed version of the Theodorsen function for closed-loop control on a pitching airfoil.

We apologise if our comment came across this way. We fully agree that a fully data-driven approach is not the only solution, it was merely our preferred choice based on the arguments given.

Unfortunately, we cannot find any reference in this direction on lines 362-365 or anywhere else in the manuscript and assume this is in reference to our comment in the responses to the author document. Therefore, we have made no modifications to the manuscript.

Minor suggestions

- Line 411: “Would have greater actuation costs”.
- Line 427: I am not entirely sure that dynamic stall time delays are fully independent of Reynolds number, particularly for utility-scale turbine blades. See Kiefer et al. (2022), Section 7.
- Line 505: Re_c is not constant in a VAWT, since it depends on the inflow velocity as well as the tangential velocity of the blade. I believe this should be presented as a range.
- Line 583: “Fourier”.

Thank you for pointing these out, they have been corrected.